# BUZz: BUffer Zones for defending adversarial examples in image classification

## Abstract

We propose a novel defense against all existing gradient based adversarial attacks on deep neural networks for image classification problems. Our defense is based on a combination of deep neural networks and simple image transformations. While straight forward in implementation, this defense yields a unique security property which we term buffer zones. In this paper, we formalize the concept of buffer zones. We argue that our defense based on buffer zones is secure against state-of-the-art black box attacks. We are able to achieve this security even when the adversary has access to the *entire* original training data set and unlimited query access to the defense. We verify our security claims through experimentation using FashionMNIST, CIFAR-10 and CIFAR-100. We demonstrate $< 10\%$ attack success rate – significantly lower than what other well-known defenses offer – at only a price of a 15-20% drop in clean accuracy. By using a new intuitive metric we explain why this trade-off offers a significant improvement over prior work.

## 1 Introduction

There are many applications based on Convolution Neural Networks (CNNs) such as image classification (Krizhevsky et al. (2012); Simonyan & Zisserman (2015)), object detection (Girshick (2015); Ren et al. (2015)), semantic segmentation (Shelhamer et al. (2017)) and visual concept discovery (Wang et al. (2017)). However, it is well-known that CNNs are highly susceptible to small perturbations which are added to *benign* input images. As shown in (Szegedy et al. (2013); Goodfellow et al. (2014)), by adding *visually imperceptible* perturbations to the original image, adversarial examples can be created. These adversarial examples are misclassifed by the CNN network with high confidence. Hence, making CNNs secure against this type of attack is a significantly important task.

The strength of an attack is relative to the considered adversarial model. In the white-box setting parameters of the target/defense model $f$ are given to the adversary – white-box attacks use this information to model access to an oracle that returns gradients $\nabla f(\cdot)$. From our analysis of previous literature, it is clear that a secure white-box defense is extremely challenging to design. We can also question the realism of having an adversary that knows the model's weights and architecture. Many online machine learning services by default only allow black-box access to their models and do not publish their model parameters (Papernot et al. (2017)). Therefore in this paper, we keep the classifier defense parameters secret. This disallows white-box attacks and so we focus exclusively on black-box adversaries in this paper.

In a black-box setting, the adversary may know some general features about the classifier (i.e. that a CNN is being used) and how the target model is trained. Most importantly, the adversary has query access to the target model itself and/or access to (part of the) training dataset. The adversary uses this information to train a synthetic model $g$. Using the synthetic model, adversarial examples can be created. The underlying assumption in the black-box setting is that a large percent of the adversarial examples created with the synthetic model will also fool the target model $f$.

Our proposed defense assumes a black-box adversary with two important attributes. First, the adversary has unlimited access to an oracle (target model) which returns the final class label $F(f(x))$. Here $f(x)$ indicates the score vector enumerating confidence scores for each possible class label and $F(.)$ computes the class label with the maximum confidence score. The second important attribute our adversary has is access to the *entire* original training dataset. In this sense, we model the strongest known black-box adversary which has not yet been studied in the literature.

**Defense based on Buffer Zones.** We first explain the concept of buffer zones. Next, we argue how "wide" buffer zones force the black-box adversary to produce a "sufficient large" noise $\eta$: As

discussed in Papernot et al. (2016a) we count an attack successful only if the adversarial noise $\eta$ has small magnitude, say $\|\eta\| \leq \epsilon$, which cannot be recognized by human beings. Forcing noise $\|\eta\| > \epsilon$ accomplishes our security goal as either the human eye or our defense detects $\eta$.

Figure 1.a describes a 2D snapshot of the landscape of a normal classifier. Three different regions with class labels $A$, $B$ and $C$ are depicted. Clearly, for any image $x$ which is close to the boundaries between the regions, the adversary can produce an adversarial example $x'$ by adding a small noise $\eta$ to $x$. The resulting adversarial image is what we would consider a true adversarial image. Here we say true in the sense that the difference between $x$ and $x'$ is almost visually imperceptible to humans, but it makes the classifier produce a wrong label.

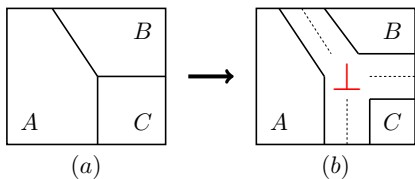

Figure 1: (a) Landscape without buffer zone, and (b) Landscape with buffer zone $\perp$.

To force the adversary to use a large perturbation $\eta$, we create buffer zones $\perp$ between all the regions as presented in Figure 1.b. Now the classifier outputs one of the class labels (A, B, or C) or $\perp$, where $\perp$ means "no class is assigned". We can think of $\perp$ as the class label of the buffer zones. In other words, the adversarial noise $\eta$ (i.e., perturbation $\eta = x' - x$) must cross over the buffer zones $\perp$ in order to modify label $l$ to label $l'$. To cross from $A$ to $B$ in Figure 1.b is initially not visible by the human eye as long as one remains in the original $A$-region of Figure 1.a and starts to transition into the $B$-region of Figure 1.a. However, to cross over into the smaller $B$-region in Figure 1.b we need an additional perturbation which we expect will become visible.

Since we only focus on small noise which is not recognized by human beings, the buffer zones $\perp$ allow us to defend against adversarial images with small noise. Only large adversarial noise, which is out of the interest of the attacker as it can be detected by the human eye, can possibly fool our defense. If the buffer zones are wide, then we accomplish our security goal since small adversarial noise cannot cross over the buffer zones, however, the prediction/clean accuracy of the defense will decrease (as it becomes more noise intolerant to both clean and adversarial examples). Notice that *it is well-known in the security community that to provide security to any system, there is an associated cost*; we believe the buffer zone concept is reasonable as experiments for our proposed techniques for creating mostly wide buffer zones show that the clean accuracy suffers only about 15-20% while reducing the attacker's success rate to about 5-10% (a 90% drop in success). Being able to reach such small attacker's success rates is a main contribution of our paper.

**Performance of a defense.** We introduce a new metric to properly understand the combined effect of (1) a drop $\gamma$ in clean accuracy from an original clean accuracy $p$ to clean accuracy $p_d = p - \gamma$ for the defense and (2) a small attacker's success rate $\alpha$ against the defense. The attacker's success rate is defined as the fraction of adversarial examples that produce an adversarial label for images that are properly classified by the defense. So, in a non-malicious environment we have the original clean accuracy $p$ while in the malicious environment the probability of proper/accurate classification by the defense model is $(p-\gamma)(1-\alpha)$ (since the defense properly labels a fraction $p-\gamma$ if no adversary is present and out of these images a fraction $\alpha$ is successfully attacked if an adversary is present). We call $(p - \gamma)(1 - \alpha)$ the *effective* clean accuracy of the defense. Going from a non-malicious environment to a malicious environment with defense gives a drop in the effective clean accuracy of[1]

$$\delta = p - (p-\gamma)(1-\alpha) = \gamma + (p-\gamma)\alpha. \qquad (1)$$

---

[1]As an example of the usefulness of $\delta$ suppose one wishes to classify a new object by taking say $n$ images and submit these images to a registration service which implements a classifier with defense. In a malicious environment the camera which is used for taking pictures or any man-in-the-middle can attempt to transform the images into adversarial examples (with a targeted new label) which cannot be detected by the human eye. Nevertheless, the service will see agreement among the produced labels of (in expectation) $(p - \delta)n$ images that are correct labeled with in the worst case the remaining images having the adversarial targeted label. This is a drop of $\delta n$ compared to a registration of an object without adversary. The smaller $\delta$, the better the defense. In order to trust a majority vote among the $n$ image labels we need $p - \delta > 0.5$.

In order to minimize this drop $\delta$, it turns out to be very important to have $\alpha$ small enough, which is accomplished in this paper. We use subscripts $t$ and $u$ in $\delta_t$ and $\delta_u$ when differentiating for targeted attacks and untargeted attacks (since untargeted attacks are easier to pull off, $\delta_t \leq \delta_u$).

**Contributions.** (a) We focus on state-of-the-art (pure and oracle-based) black box attacks where we strengthen the adversary in that we allow the adversary access to the *entire* original training data with query access to the target classifier. Related work on defense strategies is detailed in Supplemental Material B, where we argue and/or experimentally demonstrate into what extent this strong adversary can successfully attack well-known defense strategies; a summary is given in Table 1. These defenses typically (except Tramèr et al. (2017)) have an attacker's success rate of $\geq 50\%$ with a relative small drop in clean accuracy of $< 0.05$ with often almost no (close to zero percent) drop, for data sets such as MNIST and CIFAR-10. Precise calculations show that this corresponds to $\delta$ values of at least 0.29 (Tramèr et al. (2017)), 0.43 (Srisakaokul et al. (2018)), and $\geq 0.65$ for the other defenses. It turns out that for our strong adversary a vanilla network as defense – i.e., we implement no defense at all against the black-box attacker – has $\delta = 0.63$ for CIFAR-10. The other considered defenses for CIFAR-10 do not do better than implementing no defense at all.

| Defense | Data set | Attack | Att. success rate | Or. Accuracy | Def. Accuracy | $\delta$ |
|---|---|---|---|---|---|---|
| **BUZz** | CIFAR-10 | Mixed BB - iFGSM | 6.9% (this paper) | 88.35% (CIFAR-10) | 68.32% (CIFAR-10) | **0.247** |
| (Tramèr et al. (2017)) | CIFAR-10 | Mixed BB -FGSM | 34% (this paper) | 85% (CIFAR-10) | 85% (CIFAR-10) | 0.29 |
| (Srisakaokul et al. (2018)) | CIFAR-10 | Mixed BB - iFGSM | 50% (this paper) | 86% (CIFAR-10) | 86% (CIFAR-10) | 0.43 |
| **No defense** | CIFAR-10 | Mixed BB - iFGSM | 72% (this paper) | 88% (CIFAR-10) | 88% (CIFAR-10) | **0.63** |
| (Guo et al. (2017)) | CIFAR-10 | Mixed BB - iFGSM | 77% (this paper) | 84% (CIFAR-10) | 84% (CIFAR-10) | 0.65 |
| (Feinman et al. (2017)) | CIFAR-10 | Pure BB - C&W | $\geq 80\%$ (Carlini & Wagner (2017a)) | 82.6% (CIFAR-10) | 82.6% (CIFAR-10) | 0.661 |
| (Papernot et al. (2016a)) | MNIST | Oracle BB - FGSM | 70% (Papernot et al. (2017)) | 99.51% (MNIST) | 98.14% (MNIST) | 0.701 |
| (Xie et al. (2018)) | CIFAR-10 | Mixed BB - iFGSM | 86.5% (this paper) | 84% (CIFAR-10) | 59% (CIFAR-10) | 0.76 |
| (Metzen et al. (2017)) | MNIST | Pure BB - C&W | $\geq 84\%$ (Carlini & Wagner (2017b)) | 91.5% (MNIST) | 91.5% (MNIST) | 0.769 |
| (Meng & Chen (2017)) | MNIST & CIFAR-10 | Pure BB - C&W | $\geq 99\%$ (Carlini & Wagner (2017b)) | 90.6% (CIFAR-10) | 86.8% (CIFAR-10) | 0.897 |

Table 1: Attacker's success rate of black-box attacks for state-of-the-art defenses

(b) To the best of our knowledge, there are no papers on defenses against adversarial machine learning based on our concept of buffer zones. This new concept offers a conceptually simple and efficient defense strategy with rigorous security argument.

(c) We realize wide buffer zones by combining multiple classifiers with a majority vote based on a threshold together with image transformations that are unique for each of the classifiers. In Section 4 we verify our security claims through experimentation using FashionMNIST, CIFAR-10 and CIFAR-100 and show for untargeted attacks a drop in clean accuracy of 0.158, 0.200, and 0.170 in return for small attacker's success rates $\alpha < 9\%$, $< 7\%$, and $< 10\%$, respectively. This gives $\delta_u$ values 0.226, 0.247, and 0.216 showing a significant improvement over prior work – it makes sense to sacrifice some clean accuracy in return for a much smaller attacker's success rate.[2] For CIFAR-10 we conclude that BUZz with $\delta = 0.247$ (and $\alpha = 7\%$) improves over Tramèr et al. (2017) with $\delta = 0.29$ (and $\alpha = 34\%$); both are far better than any other defense, and since both use complimentary techniques we expect to be able to combine both to improve $\delta$ in future work. (All the above and this observation demonstrate the usefulness of our new $\delta$-metric.)

Finally, we will make all our code, including replicated defenses and attacks, available online.

**Outline.** We give an overview of known attacks in Section 2 and a mathematical formulation of a black-box adversary is given in Supplemental Material A. Related work on white-box and black-box defenses is given in Supplemental Material B where we also analyze (by argument and experiment) white-box defenses against black-box adversaries – this benchmarks our work. In Section 3 we detail BUZz, our defense based on buffer zones. Section 4 has experiments and simulations showing the attacker's success rates versus clean rates. Supplemental Material C enumerates pseudo code for the implemented attacks and more experimental results are given in Supplemental Material D. We conclude in Section 5.

## 2    ATTACKS

**Adversarial Examples in an image classification task.** See (Yuan et al. (2017)), the general scheme of a successful attack can be described as follows. The adversary is given a trained im-

---

[2]BUZz does better for targeted attacks with attacker's success rates $\alpha < 3.5\%$, $< 5\%$, and $< 7\%$ leading to $\delta_t$ values 0.144, 0.161, and 0.04. Here, in order to achieve $\delta_t = 0.0423$ for CIFAR-100 it turns out to be best not to implement any defense (no BUZz) – the data set must have made it already hard for a successful targeted state-of-the-art black-box attack!

age classifier (e.g, CNN network) $f$ which outputs a class label $l$ for a given input data (i.e., image) $x$. The adversary will add a perturbation $\eta$ to the original input $x$ to get an adversarial example (or a modified data input) $x'$, i.e., $x' = x + \eta$. Normally, $\eta$ should be small to make the adversarial example barely recognizable by humans. Yet, the adversary may be able to fool the classifier $f$ to produce any class label $l'(\neq l)$ as she wants. Assume that $f(x) = (s_1, s_2, \ldots, s_k)$ is a $k$-dimensional vector of confidence scores $s_j$ of class labels $j$. We call $f(x)$ the score vector with $0 \leq s_j \leq 1$, $\sum_{j=1}^{k} s_j = 1$, and $k$ the number of class labels. The class label $l$ is computed as

$$l = F(f(x)) = \operatorname*{argmax}_{i \in [1, \ldots, k]} \{s_1, s_2, \ldots, s_k\}.$$

Given $x \in [0, 1]^d$ and $l' \neq l = F(f(x))$, the attacker wishes to ideally solve

$$\min_{x' \in [0,1]^d} \|x' - x\| \text{ such that } F(f(x')) = l' \neq l = F(f(x)), \tag{2}$$

where $l$ and $l'$ are the output label of $x$ and $x'$, $\|\cdot\|$ denotes the distance between two data samples, and $d$ is the number of dimensions of $x$. $\eta = x' - x$ is the perturbation added on $x$. In this optimization problem, we minimize the perturbation $\eta$ while the label $l'$ is fixed (this represents a targeted attack). This problem becomes easier when the attacker has more information about $f(\cdot)$: In some adversarial models, the adversary may know parameters/weights that define the target model $f$. Some classification applications may directly output vector $f(x)$ instead of $F(f(x))$ and this gives more information about the target model.

By adding a sufficient large noise $\eta$ to any given *benign* input image, we can fool any existing image classifier. However, as discussed in (Papernot et al. (2016a)) we count an attack successful only if the adversarial noise $\eta$ has small magnitude, say $\|\eta\| \leq \epsilon$, which cannot be recognized by human beings ($\epsilon$ indicates this transition from noise not being recognized to noise that is visually perceptible). In this sense the attacker is already successful as soon as a sufficiently small perturbation $\eta$ ($\|\eta\| \leq \epsilon$) is found that realizes label $l'$. That is, finding the *minimal* possible perturbation $\eta$ in (2) is not necessary.

**Attack Methodologies.** Attacks can be partitioned into two main categories based on their approach. The first kind are white-box attacks where the adversary knows the parameters of defense/target model/classifier $f$ and uses these parameters to compute gradients. One can think of this scenario as having oracle access to gradients $\nabla f(x)$ for input images $x$. The attacker only uses this type of oracle access to compute adversarial examples.

The second kind are black-box attacks where the adversary does not know the parameters of $f$, but does have black-box access to the target model itself. One can think of this scenario as having oracle access to class labels $F(f(x))$ or score vector values $f(x)$ (the latter gives more information and models a stronger attacker). In addition to having black-box access to the target model, the adversary may know and use (part of) the original training data (this can be used to train an adversarial synthetic model which resembles the target model). Since the oracles given to the white-box and black-box attackers are different/complimentary, white-box defenses and black-box defenses deal with different attack methodologies.

**White-box Attacks.** (Yuan et al. (2017)) constructs perturbation $\eta$ with the help of gradient $\nabla f(\cdot)$: for example, $\eta = \epsilon \times sign(\nabla_x L(x, l; \theta)$ in the Fast Gradient Sign Method (FGSM) by (Goodfellow et al. (2014)), where $\theta$ represents the parameters of $f$, $L$ is a loss function (e.g, cross entropy) of model $f$. ($\epsilon$ can be thought of as relating to the maximum amount of noise which is not visually perceptible.)

**Black-box Attacks.** Black-box attacks use non-gradient information of classifier $f$ such as (part of) the original training data set $\mathcal{X}_0$ (Papernot et al. (2016b)) and/or a set $\mathcal{X}_1$ of adaptively chosen queries to $f$ (i.e., $\{(x, f(x)) : x \in \mathcal{X}_1\}$ or $\{(x, l = F(f(x))) : x \in \mathcal{X}_1\}$) (Papernot et al. (2017)) – querries in $\mathcal{X}_1$ are not in the training data set $\mathcal{X}_0$. These type of attacks exploit the *transferability* property of adversarial examples (Papernot et al. (2016b); Liu et al. (2017)): Based on information $\mathcal{X}_0$ and $\mathcal{X}_1$ the adversary trains its own copy of the proposed defense. This is called the adversarial synthetic network/model and is used to create adversarial examples for the target model. (Liu et al. (2017)) shows that the transferability property of adversarial examples between different models which have the same topology/architecture and are trained over the same dataset is very high, i.e., nearly $100\%$ for ImageNet (Russakovsky et al. (2015)). This explains why the adversarial examples

generated for the synthetic network can often be successful adversarial examples for the defense network.

Black-box attacks can be partitioned into three following categories:

- **Pure black-box attack** (Szegedy et al. (2014); Papernot et al. (2016b); Athalye et al. (2018a); Liu et al. (2017)). The adversary is *only* given knowledge of a training data set $\mathcal{X}_0$. Based on this information, the adversary builds his own classifier $g$ which is used to produce adversarial examples using an existing white-box attack methodology. These adversarial examples of $g$ may also be the adversarial examples of $f$ due to the transferability property between $f$ and $g$.

- **Oracle based black-box attack** (Papernot et al. (2017)). The adversary is allowed to adaptively query target classifier $f$, which gives information $\mathcal{X}_1$. Based on this information, the adversary builds his own classifier $g$ which is used to produce adversarial examples using an existing white-box attack methodology. Again, the generated adversarial examples for $g$ may also be able fool classifier $f$ due to the transferability property between $f$ and $g$. Compared to the native (pure) black box attack, this attack is supposed to be more efficient because $g$ is intentionally trained to be similar to $f$. Hence, the transferability between $f$ and $g$ may be significantly higher.

- **Zeroth Order Optimization based black-box attack** (Chen et al. (2017)). The adversary does not build any assistant classifier $g$ as done in the previous black-box attacks. Instead, the adversary adaptively queries $\{x, f(x), F(f(x))\}$ to approximate the gradient $\nabla f$ based on a derivative-free optimization approach. Using the approximated $\nabla f$, the adversary can build adversarial examples by directly working with the classifier $f$.

In this paper, we analyze a **mixed black-box attack** where the synthetic network $g$ is built based on the training data set $\mathcal{X}_0$ of the target model $f$ and is based on adaptively chosen queries $\mathcal{X}_1$. Our mixed black-box attack is more powerful than both the pure black-box attack and oracle based black-box attack. Supplemental material C provides pseudo code.

As explained and motivated in the introduction, we restrict ourselves to the black-box setting where we keep secret the parameters of our defense classifier, called BUZz (see next section), and we do not reveal score vectors – this disallows white-box attacks and zeroth order optimization based black-box attacks.

## 3 BUZz: A defense based on buffer zones

**Design Philosophy.** Since each single network gives a classifier with aligned boundaries (i.e. no buffer zones), we propose to combine multiple classifiers (each with its own aligned boundaries) to produce a composed classifier which will provide a non-empty buffer zone. To create a buffer zone we use majority voting among the individual classifiers with a threshold mechanism. E.g., the threshold may be such that only if all individual classifiers agree on the same label $l$, then the composed classifier outputs $l$, otherwise, it outputs $\perp$. Because the transferability among all the individual classifiers is not perfect, they will disagree if an image has 'too much' noise and this leads to an output $\perp$. The area where they disagree is the buffer zone.[3]

Although majority voting based on a threshold leads to the existence of a buffer zone, the resulting buffer zone may not be wide enough to prevent a successful attack using small adversarial noise. If we are able to decrease/diminish the transferability among the different classifiers, then this leads to a wider buffer zone. To decrease the transferability, we must make the individual classifiers more unique. This can be done by, for each CNN network, first uniquely transforming the inputted image $x$. Since the transformations are different for each of the classifiers that participate in the majority

---

[3] The buffer zone concept offers a first immediate insight: A defense with only one single classifier with buffer zones may be hard to develop because of its nature – a single network does not produce buffer zones between regions unless (1) another class label $\perp$ for the buffer zone can be trained, but how does one construct proper training data? or (2) the score vector is used to mathematically define a subspace of score vectors that should map to $\perp$, but how can one achieve acceptable clean accuracy at the same time? For this reason our technique for creating a non-empty buffer zone uses multiple classifiers.

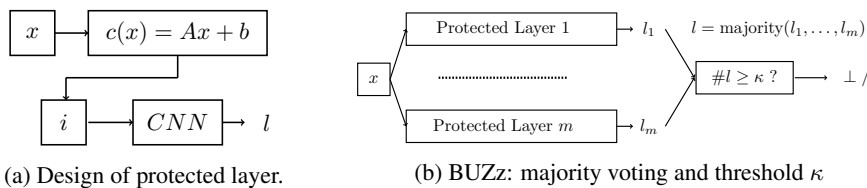

(a) Design of protected layer.   (b) BUZz: majority voting and threshold $\kappa$

Figure 2: General Design of BUZz.

vote, transferability will be reduced. We will discuss what kind of transformations reduce the clean accuracy the least.

The created buffer zones may be mostly wide, but there will exist narrow parts. Since the narrow parts allow a successful attack based on small adversarial noise, the attacker will want to search for a narrow part around a given image $x$. In order to make such a search less efficient and less often successful, the distinct transformations are kept secret. That is, only the distribution (or set) from which the transformations are drawn is known to the adversary (the concrete transformation selections/parameters are kept secret). We will make the adversarial model explicit once we have detailed our defense mechanism:

**BUZz using multiple classifiers and secret image transformations.** Our defense is composed of multiple CNNs as depicted in Figure 2b. Each CNN has two *unique image transformations* as shown in Figure 2a. The first is a fixed randomized linear transformation $c(x) = Ax + b$, where $A$ is a matrix and $b$ is a vector. After the linear transformation a resizing operation $i$ is applied to the image before it is fed into the CNN. The CNN corresponding to $c$ and $i$ is trained on clean data $\{i(c(x))\}$. This results in a weight vector $w$. The $m$ protected layers in Figure 2b are described by 'parameters' $(c_j, i_j, w_j)_{j=1}^m$.

When a user wants to query the defense, input $x$ is submitted to each layer which computes its corresponding image transformation and executes its CNN. The outputs of the layers are class labels $(l_j)_{j=1}^m$. The final class label of BUZz, i.e., the composition of the $m$ protected layers, is a majority vote based on a threshold $\kappa$. In our experiments we use unanimous voting, i.e., if the networks do not all output the same class label then the adversarial/undetermined class label $\perp$ is given as the output (i.e., $\kappa = m$).

**Adversarial Model.** In our adversarial model we assume that no score vectors are revealed (which makes the attacker incapable of executing a zeroth order optimization based black-box attack) and we assume that the parameters $(c_j, i_j, w_j)_{j=1}^m$ are kept secret, i.e., the attacker has no direct knowledge about the weights of the CNN networks, matrices $A$ and vectors $b$, and the amount of image resizing for each layer (this makes the attacker incapable of executing a white-box attack).

Our adversarial model allows a mixed black-box attacker having more capabilities (is stronger) than the strongest black-box adversary in literature: Just like in (Papernot et al. (2017)), the adversary is allowed to query the defense as many times as they desire and is practically possible, they may generate synthetic data $\mathcal{X}_1$ and they can train a synthetic network. Based on this synthetic network they can carry out white-box attacks and test their efficiency (attack success rate) on the defense. We give the adversary one additional and extremely powerful ability that (Papernot et al. (2017)) does not. We allow the adversary access to the *entire* original training data set $\mathcal{X}_0$ as an initial set. This gives the adversary access to a huge amount of training data, an order of magnitude higher than what (Papernot et al. (2017)) gives in the original attack.

In supplemental material A we mathematically formalize the adversary (as is done in crypto/security literature) as an adversarial algorithm in order to make precise the adversary's capabilities.

**Image Transformations.** From (Guo et al. (2017)) we know adversarial examples are sensitive to image transformations which either distort the value of the pixels in the image or change the original spatial location of the pixels. However, it is well established (Goodfellow et al. (2016)) that CNNs rely on certain spatial patterns being present in the data in order to classify clean (non-adversarial) data. Hence, we want an image transformation that keeps such patterns 'invariant' while introducing distortions that make the attacker's task less likely to succeed.

In the literature, previous defenses with only a single network have suggested multiple different image transformations (Meng & Chen (2017); Xie et al. (2018)). Through experimentation we decided to employ two image transformations for each protected layer. The first transformation is a simple resizing operation, where we resize the image before giving it as input to the CNN. Resizing to a smaller dimension than the original image may result in lost pixel information and by extension hurt the network's performance on clean data. Therefore we only consider resizing operations which increase the size of the image. The other transformation we use is a linear transformation: $c(x) = Ax + b$ where $x$ is an $n$ by $n$ pixels input image, $A$ is a fixed $n$ by $n$ matrix and $b$ a fixed $n$ by $n$ pixels 'noise' sample. Depending on the data set we can control the trade off between the attacker's success rate and clean accuracy using the linear transformation. For example if only $b$ is random (with small magnitude) and $A$ is identity, it results in less image distortion (so higher clean accuracy) but also less security (more adversarial samples bypass the defense).[4]

**Security Argument.** In the context of Papernot's oracle based black-box attack and pure black-box attack, the adversarial noise $\eta$ is created based on a white-box attack for a synthetic network $g$ of BUZz. It means that the noise $\eta$ is specifically developed for $g$. Since the $x' = x + \eta$ is inputted into every protected layer of BUZz, the $j$-th layer will apply its CNN network on a noisy image $i_j(c_j(x')) = i_j(c_j(x + \eta))$, which due to the linearity of $i_j(c_j(\cdot))$ is equal to $i_j(c_j(x)) + i_j(c_j(\eta))$. Therefore, layers receive different inputted noises $i_j(c_j(\eta)) \neq \eta$. Hence, the protected layers have different behavior from one another and from synthetic network $g$ for any given adversarial example $x' = x + \eta$. This widens the buffer zones as it is less likely that each protected layer reacts the same to $\eta$ in terms of miss-classification.

## 4 EXPERIMENTAL RESULTS

In this section we discuss our experiments for CIFAR-10; Supplemental Material D has also experiments for FashionMNIST and CIFAR-100. Supplemental material C provides exact details in the form of pseudo codes and tables with attack parameters; below is a concise summary in words.

We implemented the mixed black-box attack, i.e., the state-of-the-art oracle based black-box attack of Papernot et al. (2017) where in addition the adversary has access to the entire training data set. The attack first generates a synthetic network by taking the initial training data set $\mathcal{X}_0$ and learning the parameters $\theta_g$ of a single vanilla network $g$ – for the training we use Adam (Kingma & Ba (2014)) with learning rate 0.0001, batch size 64, 100 epochs, and no data augmentation. During the first iteration the Jacobian matrix of the score vector $g$ is computed in each image $x \in \mathcal{X}_0$. The signs of the entries in the Jacobian matrix that correspond to $x$'s class label (according to the target model) form a vector which is scaled with $\lambda = 0.1$ and added to $x$. This leads to an augmented data set $\mathcal{X}_1$ which consists of these new images together with $\mathcal{X}_0$. We use black-box access to the target model to learn the labels for the images in $\mathcal{X}_1$. During next iterations we double the data set 5 times and learn a more accurate synthetic network $g$.

Second, the attacker uses Carlini's single synthetic network $g$ (Carlini & Wagner (2016)) and applies a targeted/non-targeted iterative/non-iterative FGSM attack (we use the cleverhans library, see `https://github.com/tensorflow/cleverhans`) to produce an adversarial example $x'$ for image $x$. (We do not use the C&W attack (Carlini & Wagner (2016)) because our experiments show a much lower attacker's success rate.) During each FGSM iteration the black-box adversary has the capability to use black-box access to the target model to verify whether the produced $x'$ in that iteration has a desired label $l'$. We use 10 iterations in iterative FGSM with $\epsilon = 10/256$ (giving a $1/256$ scaling vector in each of the 10 iterations). In non-iterative FGSM we pick $\epsilon = 1/20$.

The CNN networks in the vanilla network and each of the protected layers of BUZz (the target model) each have the VGG16 architecture (see `https://neurohive.io/en/popular-networks/vgg16/8`) and are trained on clean data $\{(i_j(c_j(x)), l)\}$, where $l$ is the to-be-learned class label for $x$. BUZz uses anonymous voting with $\kappa$ equal to the number of networks. The image transformations are selected from mappings $c(x) = x + b$ (we use the identity matrix for $A$) where the entries of $b$ are non-zero with probability $p = 0.35$ and non-zero entries

---

[4]We note that the positive results reported in this paper open the door for the exploration of many more possible image transformations. Other transformations such as the affine transformation, zero padding images and pixel shuffling could yield even better trade off between security and clean accuracy in multi model defenses. We leave these possibilities as future work.

Table 2: Mixed black-box attack on CIFAR-10

|  | Vanilla | 2-Networks | 4-Networks | 8-Networks |
|---|---|---|---|---|
| Clean Accuracy | 0.8835 | 0.7599 | 0.6832 | 0.6091 |
| FGSM Targeted | 0.831 | 0.951 | 0.985 | 0.995 |
| iFGSM Targeted | 0.766 | 0.978 | 0.998 | 1 |
| FGSM Untargeted | 0.297 | 0.797 | 0.931 | 0.976 |
| iFGSM Untargeted | 0.282 | 0.814 | 0.963 | 0.992 |

Table 3: $\delta$ values for mixed black-box attacks on FashionMNIST, CIFAR-10, and CIFAR-100

|  | Defense targeted | $\delta_t$ | Defense untargeted | $\delta_u$ |
|---|---|---|---|---|
| FashionMNIST | 4-Networks | 0.144 | 8-Networks | 0.226 |
| CIFAR-10 | 2-Networks | 0.161 | 4-Networks | 0.247 |
| CIFAR-100 | Vanilla | 0.0423 | 2-Networks | 0.216 |

are uniformly selected from the interval $[-0.5, +0.5]$. Images $x$ are $32 \times 32 \times 3$ where 3 stands for the red, blue, green values (each in $[-0.5, +0.5]$) of a pixel. The same non-zero entry of $b$ is added to each of the red, blue, green values of the corresponding pixel in $x$. (We can think of an image transformation $c_j(x)$ as an extra randomly fixed layer added to the layers which form the $j$-th CNN.) We tested three BUZz designs: One with 8 networks each using a different image resizing operation from 32 to 32, 40, 48, 64, 72, 80, 96, 104. The second with 4 networks being the subset of the 8 networks that use image resizing operations from 32 to 32, 48, 72, 96. The third with 2 networks being a subset of the 8 networks that use image resizing operations from 32 to 32 and 104.

Table 2 depicts the results of the four possible target/untargeted iterative/non-iterative FGSM attacks against a plain vanilla network without image transformation and BUZz with 2, 4, and 8 networks respectively. The *defense rate* is defined as $1 - \alpha$, where $\alpha$ is the attacker success rate computed as the fraction of 1000 test data (with the property that the target model produces the correct label) for which the adversary produces a successful adversarial example $x'$ that changes the label to a desired label $l'$. The clean rate and defense rate[5] are an average over 5 runs where each run anew selects random image transformations in BUZz. The experiments show that the min and max values over runs are not far apart indicating that most random image transformation selections according to the recipe described above give similar results.

For BUZz with 4 networks for CIFAR-10 we see a 0.2003 drop in clean accuracy from 0.8835 to 0.6832. In return for this drop in clean accuracy we see a defense rate of $\geq 0.931$, i.e., an attacker's success rate $< 7\%$. Also for FashionMNIST and CIFAR-100 (for BUZz with 8 resp. 2 networks) we see drops in clean accuracy of 0.1577 and 0.1703 in order to achieve an attacker's success rate of $< 10\%$ and $< 9\%$ (see supplemental material D). This leads to the $\delta$ values reported in the introduction. Table 3 shows the best BUZz configurations with targeted and untargeted $\delta$ values.

## 5 CONCLUSION

We introduced a new concept called buffer zones which is at the core of a new adversarial ML defense, coined BUZZ. BUZz defends against black-box adversaries with oracle access to the target model (BUZz) and knowledge of the entire training data set. BUZz uses threshold voting over multiple networks that each are preceded with a secret/hidden image transformation. Experiments for FashionMNIST, CIFAR-10, and CIFAR-100 for carefully designed classes of image transformations in BUZz show that at the cost of drop in clean accuracy of 15-20% the attacker's success rate is only 5-10% – much less than the best attacker's success rates $\geq 34\%$ achieved by other well known defenses for these data sets. We have argued this to be an acceptable trade-off for better security by using a new metric (called $\delta$ value).

---

[5]One would expect a lower defense rate for iFGSM attacks compared to FGSM attacks. This is not reflected in the table due to the choices of $\epsilon$ taken from literature which are different for iFGSM and FGSM. For future work we anticipate a detailed sensitive study with respect to $\epsilon$.

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

# A  ADVERSARIAL MODEL

The strength of an attack is relative to the considered adversarial model. In adversarial Machine Learning (ML) the assumed capabilities of an attacker are a mix of:

**Having knowledge of the parameters and architecture of the defense network/classifier.** The architecture and methodology of how the defense network is trained is about how the defense operates and its underlying philosophy. In cryptography this is similar to the actual design of a security primitive and this is never assumed to be hidden as this would lead to the undesirable practice of security by obscurity. The secret key of the security primitive is kept private; the method of how the secrect key is generated using a probabilistic algorithm is public. Similarly, the parameters of a defense network can be considered to be private while the method of their training is public. If the adversarial model does not assume the parameters to be private, then all is public and we call this a **white-box setting**. If the parameters are considered to be private and not given to the attacker (for 'free'), then we call this a **black-box setting**.

**Having access to the defense classifier a.k.a. target model.** If the parameters are kept private, then the next best adversarial capability is having access to the target model as a black-box. The idea is that the adversary can adaptively query the target model with own input images for which class labels are being returned. Here, we have two flavors: only a class label is returned, or more information is given in the form of a score vector which entries represent confidences scores of each of the classes (the returned class label is the one with maximal confidence score).

In the white-box setting where all parameters are known, the attacker can reproduce the target model and access the target model as a black-box. Confusing in adversarial ML is that **white-box attacks** are the ones that 'only' use the parameters in the white-box setting to learn gradients of the target model which are used to produce adversarial examples – these attacks do not consider/use black-box access. This means that white-box defenses are not necessarily analysed against **black-box attacks** where the adversary only has black-box access to the target model with possibly the added capability described below.

**Having access to (part of the) training data.** The training data which is used to train the parameters of the defense network is often public knowledge. Knowing the methodology of how the defense network is trained, an adversary can apply the same methodology to train its own synthetic defense network – and this can be used to find adversarial examples. The synthetic network will not be exactly the same as the defense network since training is done by randomized (often SGD-like) algorithms where training data is used in random order. This means that knowledge of the training data is less informative than knowledge of the parameters as in the white-box setting.

The white-box setting describes the capabilities of the strongest adversary, while the black-box setting describes a weaker adversary who cannot exactly reproduce the target model (and can only estimate the target model by training a synthetic network). White-box attacks on the other hand restrict the adversary in that only oracle access to the gradient of $f$ is allowed. Black-box attacks only allow oracle access to the target model itself and oracle access to training data. In this sense white-box attacks in adversarial ML literature exclude access to the above black-box oracles. This means that even though a white-box defense may be able to resist a white-box attack, it can still be vulnerable to a black-box attack. Vice versa, even though a white-box defense may be broken under a white-box attack, it may still survive in the black-box setting.

Taking BUZz as an example, we may mathematically formalize the black-box adversary[6] (as is done in crypto/security literature) as an adversarial algorithm $\mathcal{A}_T$ which has access to

- a random oracle representing $l \leftarrow \text{BUZz}_\theta(x)$ where parameters $\theta = (c_j, i_j, w_j)_{j=1}^m$, input $x$ is an image, and $l$ is the outputted label of the target classifier BUZz (the collected $(x, l)$ is the set $\mathcal{X}_1$ of Section 2), and

---

[6]Similarly a white-box adversary with only oracle access to gradient information can be modeled.

- a random oracle $\xi_D$ which outputs at most $D$ times 'training data' according to the distribution from which the training data is taken from (the collected $(x, l)$ is the set $\mathcal{X}_0$ of Section 2); by abuse of notation $\xi$ denotes the distribution itself.

Subscript $T$ denotes the allowed number of computation steps plus oracle accesses to $\text{BUZz}_\theta$ and $\xi_D$. In our experiments we test the most powerful existing black-box attacks and do not mention $T$; here, $T$ just means the amount of steps allowed by existing practical attack methodologies. If an attacker with unlimited access ($T = \infty$) to $\text{BUZz}_\theta$ can scan region boundaries, then this achieves optimal success rates.

Subscript $D$ in $\xi_D$ indicates the number of training data an attacker is allowed to use for the attack. $D$ represents an important metric in machine learning as the amount of training data cannot be assumed infinite (with respect to the application there is a concrete limit to how many training data is available; collecting samples from $\xi$ is not straightforward, e.g., making a true new picture/image of a plane takes effort). How strong we make e.g. Papernot's black-box attack is based on how much training data we give it: In their paper $D = 150$ is used for MNIST in order to train a synthetic network while in our experiments we use $D = 50K$ which is the *entire* original training data set of CIFAR-10 (and leaves 10K test data). Since the attack uses the synthetic network in a (targeted or untargeted) white-box attack (with small enough $\epsilon$) to generate an adversarial example, the probability of successfully changing the label depends on how similar the synthetic network classifies data compared to the target model with defense – it depends on the tranferability between the synthetic and defense classifiers and transferability improves for larger $D$.

The aim of the adversary is to produce a perturbation $\eta \leftarrow \mathcal{A}_T^{\xi_D, \text{BUZz}_\theta}(x)$ (just based on the oracle accesses described above indicated by superscripts and based on input $x$) which is visually imperceptible, i.e. $\|\eta\| \leq \epsilon$, and for which $x' = x + \eta$ gives a different label: The attacker's success rate for untargeted black-box attacks is defined as the probability

$$Pr_{x \leftarrow \xi}(\eta \leftarrow \mathcal{A}_T^{\xi_D, \text{BUZz}_\theta}(x), \ \|\eta\| \leq \epsilon, \ \text{BUZz}_\theta(x + \eta) \notin \{\bot, \text{BUZz}_\theta(x)\} \mid \text{BUZz}_\theta(x) \neq \bot).$$

For targeted black-box attacks we replace $\text{BUZz}_\theta(x + \eta) \notin \{\bot, \text{BUZz}_\theta(x)\}$ by $\text{BUZz}_\theta(x + \eta) = l'$, replace $\text{BUZz}_\theta(x) \neq \bot$ by $\text{BUZz}_\theta(x) \notin \{\bot, l'\}$, and take the probability over both $x \leftarrow \xi$ and $l' \leftarrow [1..k]$. In the above notations we do not explicitly state that the adversary also has knowledge of the distribution from which $\theta$ is taken, i.e., the adversary knows the philosophy behind our defense together with what type of image transformations are being used and knows the architecture in terms of number of nodes and connections at each layer of the CNN networks and how they are trained.

The above formalism helps in making the adversarial model in terms of adversarial capabilities precise. We will not explicitly use the formalism as we cannot prove statements about general classes of adversarial algorithms $\mathcal{A}$ (our defense does not allow 'standard' reduction proofs to some hard computational problem as is done in crypto).

From a cryptographer's perspective we want the above probability (i.e., the attacker's success rate) to be very small, even negligible in some security parameter $\lambda$ where image $x$ is $poly(\lambda)$-sized. First, in ML we have concrete data sets with images of certain fixed sizes for which we want to design defense strategies against adversarial ML. So, such an asymptotic goal makes no sense. Second, it turns out that it is very difficult to obtain a defense strategy that can make the attacker's success rate very small, say 0.1%, without sacrificing the clean accuracy all together. In this paper we have been able to reduce the attacker's success rate down to about 5-10% while only reducing the clean accuracy by 15-20%.

## B  RELATED WORK: COMPARISON TO KNOWN DEFENSES

**White-Box Defenses.** White-box defenses are any defense with an adversarial model that allows the adversary oracle access to the gradient of the target model. These defenses include (Papernot et al. (2016a); Kurakin et al. (2016); Tramèr et al. (2017); Cao & Gong (2017); Metzen et al. (2017); Feinman et al. (2017); Xie et al. (2018); Meng & Chen (2017); Srisakaokul et al. (2018)) to name a few. A complete list is given in (Athalye et al. (2018a); Carlini & Wagner (2017a)) except for the unpublished work (Srisakaokul et al. (2018)) which appeared later. So far, any defense with public weights and architecture turns out to be vulnerable to FGSM, IFGSM, or Carlini type attacks (Carlini

& Wagner (2017a;b); Athalye et al. (2018a); Liu et al. (2017); we will argue the vulnerability of (Srisakaokul et al. (2018)) below).

In order to implement a white-box attack (Yuan et al. (2017)) constructs perturbation $\eta$ with the help of gradient $\nabla f(\cdot)$: for example, $\eta = \epsilon \times sign(\nabla_x L(x, l; \theta))$ in the Fast Gradient Sign Method (FGSM) by (Goodfellow et al. (2014)), where $\theta$ represents the parameters of $f$, $L$ is a loss function (e.g, cross entropy) of model $f$. ($\epsilon$ can be thought of as relating to the maximum amount of noise which is not visually perceptible.) To defend against adversarial examples, many methodologies have been proposed and they all employ the same strategy, i.e., *gradient masking* (Papernot et al. (2017)) respectively *obfuscated gradient* (Athalye et al. (2018a)). As pointed out in (Athalye et al. (2018a)), there are three main methods for realizing this strategy: *shattered gradients*, *stochastic gradients* and *exploding & vanishing gradients*. In (Athalye et al. (2018a)), the authors propose three different types of attacks:

1. **Backward Pass Differentiable Approximation (BPDA)**. The attack is applied for protected network $f(t(x))$ where $t(x)$ is not differentiable and $t(x) \approx x$. The adversary will replace $t(x)$ in the backward phase for computing the gradient by $x$ and thus, he can compute the approximated gradient $\nabla_x f(t(x))|_{x=\hat{x}} \approx \nabla_x f(x)|_{x=t(\hat{x})}$.

2. **Expectation over Transformation (EOT)**. In this case, the adversary computes the gradient of $\mathbb{E}_{t \sim T} f(t(x))$ where $t(x)$ is a random transformation and $t$ is sampled from a distribution $T$. The gradient can be computed as $\nabla \mathbb{E}_{t \sim T} f(t(x)) = \mathbb{E}_{t \sim T} \nabla f(t(x))$.

3. **Reparameterization.** The protected network $f(t(x))$ has $t(x)$ which performs some optimization loop to transform the input $x$ to a new input $\hat{x}$. This step will make the gradient exploding or vanishing, i.e., the adversary cannot compute the gradient. To cope with this defense, Athalye et al. (2018a) proposes to make a change-of-variable $x = h(z)$ for some $h(\cdot)$ such that $t(h(z)) = h(z)$ for all $z$ but $h(\cdot)$ is differentiable.

In literature, many white-box defenses have shown a predictable cat and mouse type of security game. In this repeated chain of events, the defender creates a network defense and the attacker comes up with a new type of attack that breaks the defense. The defender then creates a new defense which the attacker again breaks. While this occurs frequently in security, a simple example of this occurring in the field of adversarial machine learning is the FGSM attack breaking standard CNNs, the distillation defense mitigating FGSM, and the distillation defenses subsequent break by (Carlini (Papernot et al. (2016a); Carlini & Wagner (2017b))). Alternatively, in an even worse case, the defense can be immediately broken without the need for new attack strategies. In adversarial machine learning an example of this is the autoencoder defense of (Meng & Chen (2017)) which is vulnerable to the attack in (Carlini & Wagner (2017b)). From our analysis of the previous literature it is clear that a secure pure white-box defense is extremely challenging to design.

**Black-Box Defenses based on a Single Network.** We discuss how the white-box defenses of (Papernot et al. (2016a); Kurakin et al. (2016); Tramèr et al. (2017); Cao & Gong (2017); Metzen et al. (2017); Feinman et al. (2017); Xie et al. (2018); Meng & Chen (2017); Guo et al. (2017); Srisakaokul et al. (2018)) are vulnerable in a black-box setting. As shown in (Papernot et al. (2017)), the adversary can build a synthetic network $g$ which simulates the target vanilla network. This can be used to produce high transferability adversarial examples (that transfer to the target model with significant success). Boundary alignment is the well-known explanation, see (Papernot et al. (2016b)).

(Papernot et al. (2016a)) proposes a single network defense with a better adversarial robustness property based on distillation: First, given a training data set, a network is built and trained. After this, the softmax output (i.e., score vector) of the network is used to train another network with the original training data set. This process is called 'distillation' and the distilled network is argued to have better robustness against white-box attacks. However, (Carlini & Wagner (2016)) showed a white-box attack against this defense. Moreover, (Papernot et al. (2017)) showed that for the MNIST dataset, the success rate of Papernot's black-box attack (untargeted) is at least $70\%$.

In (Kurakin et al. (2016)), the authors discuss how to train the network for ImageNet with adversarial examples to make it robust against adversarial machine learning. During each epoch in the training process, adversarial examples are generated and again used in the training process. According to Table 4 in (Kurakin et al. (2016)), the success rate of (untargeted) pure black box attack on ImageNet using FGSM is high $\geq 50\%$. The authors in (Tramèr et al. (2017)) also claim that the adversarial training in (Kurakin et al. (2016)) may not be useful.

(Tramèr et al. (2017)) proposes another type of adversarial training method. The adversarial examples are generated by doing attacks on different networks with different attack methods. After this the designer trains the new network with the generated adversarial examples. The authors argued that this adversarial training can make the adversarially trained network more robust against (pure) black-box attacks because it is trained with adversarial examples from different sources (i.e., pre-trained networks). In other words, the network is supposed to have better robustness against black-box attack generalization across models. As shown in (Athalye et al. (2018b)), the adversarially trained network is vulnerable to white-box attack. Regarding pure black-box attack, as reported in Table 4 in (Tramèr et al. (2017)), the success rate of (untargeted) pure black box attack on ImageNet using FGSM – the best known black box attack that has been executed on this defense – is $\geq 27\%$. We verify the efficiency of this approach for CIFAR-10 in this paper. We do the adversarial retraining using data from 8 other networks to build a 9th network. The 8 other networks are from the Mul-Def paper Srisakaokul et al. (2018) (we also rigorously discuss this paper next few paragraphs). Network 0, is a vanilla VGG. Network 1 is a VGG with 30% adv training from network 0. Network 2 is a VGG with 30% adv training (15% from network 0, 15% from network 1). Etc. until we get to Network 8. After training we run the full Papernot attack on Network 8. The full result we can find in Table 4, i.e., Adv Retrained 1-Net. The defense has clean prediction accuracy of 85% and the best attack on the defense is untargeted FGSM or iFGSM with success rate of 34%.

(Cao & Gong (2017)) proposes a white-box defense based on the following trick: for a given sample $x$, the defense collects many samples $x'_1, \cdots, x'_n$ in a small hypercube centered at $x$. Then, the outputted class label is the one which gains the majority vote among $F(f(x'_1)), \cdots, F(f(x'_n))$ where $F$ is the output function of network $f$. We argue that this defense is vulnerable to black-box attacks because of the following reasons. The adversary can build a synthetic network $g$ with very high transferability between $f$ and $g$ (Papernot et al. (2016b); Liu et al. (2017)). After this, the attacker looks for adversarial examples which can gain the majority vote in the same setting as proposed above in (Cao & Gong (2017)). We believe that there exist many such adversarial examples, in particular, if the clean $x$ is very close to the decision boundary, then the adversarial example $x' = x + \eta$ can lie deeply in another region and its distance to the decision boundary is larger than that of $x$. Hence, $x'$ now gains the majority vote and fools the defense. This example also shows the importance of our buffer zone concept as a defense mechanism.

In (Metzen et al. (2017)), the authors constructed a defense of a single network $f$ with an additional 'detector' network $g$. The 'detector network' is built based on the training data set of the main network $f$ together with adversarial examples generated for training data samples. The detector network is used to distinguish clean samples from adversarial samples. The authors in (Carlini & Wagner (2017a)) showed white-box and black-box attacks on this defense. The success rate of untargeted pure black-box attack on MNIST using the C&W attack by (Carlini & Wagner (2016)) is at least 84%.

In (Feinman et al. (2017)), the authors built a detector to distinguish adversarial examples from clean examples using Bayesian uncertainty estimate or Kernel Density Estimator. The key idea is that since the adversarial and clean examples do not belong to the same manifolds, the defender can build a detector. (Carlini & Wagner (2017a)) showed a white-box attack on this defense and clearly claim that the defense does not work if the dataset is CIFAR-10 for both white-box attack and black-box attack, i.e., the success rate of untargeted pure attack seems at least 80% based on their explanation.

Table 4: Mixed black-box attacks on defenses Randomized 1-Net (Xie et al. (2018)), Adv Retrained 1-Net (Tramèr et al. (2017)), Mul-Def 2,4,8-Net (Srisakaokul et al. (2018)), Mixed Arch 2-Net (Liu et al. (2017)) and BUZz 2, 4, 8-Net.

| | Clean Accuracy | FGSM targeted | iFGSM targeted | FGSM untargeted | iFGSM untargeted |
|---|---|---|---|---|---|
| Randomized 1-Net (Xie et al. (2018)) | 0.58 | 0.83 | 0.72 | 0.32 | 0.13 |
| Adv Retrained 1-Net (Tramèr et al. (2017)) | 0.85 | 0.93 | 0.93 | 0.66 | 0.67 |
| BUZz 1-Net | 0.83 | 0.84 | 0.73 | 0.26 | 0.22 |
| Mul-Def 2-Net (Srisakaokul et al. (2018)) | 0.86 | 0.85 | 0.82 | 0.44 | 0.39 |
| Mixed Arch 2-Net (Liu et al. (2017)) | 0.71 | 0.96 | 0.98 | 0.82 | 0.87 |
| BUZz 2-Net | 0.75 | 0.95 | 0.97 | 0.82 | 0.83 |
| Mul-Def 4-Net (Srisakaokul et al. (2018)) | 0.86 | 0.90 | 0.86 | 0.54 | 0.48 |
| BUZz 4-Net | 0.68 | 0.98 | 0.99 | 0.93 | 0.96 |
| Mul-Def 8-Net (Srisakaokul et al. (2018)) | 0.85 | 0.92 | 0.87 | 0.59 | 0.50 |
| BUZz 8-Net | 0.61 | 0.99 | 1.00 | 0.97 | 0.98 |

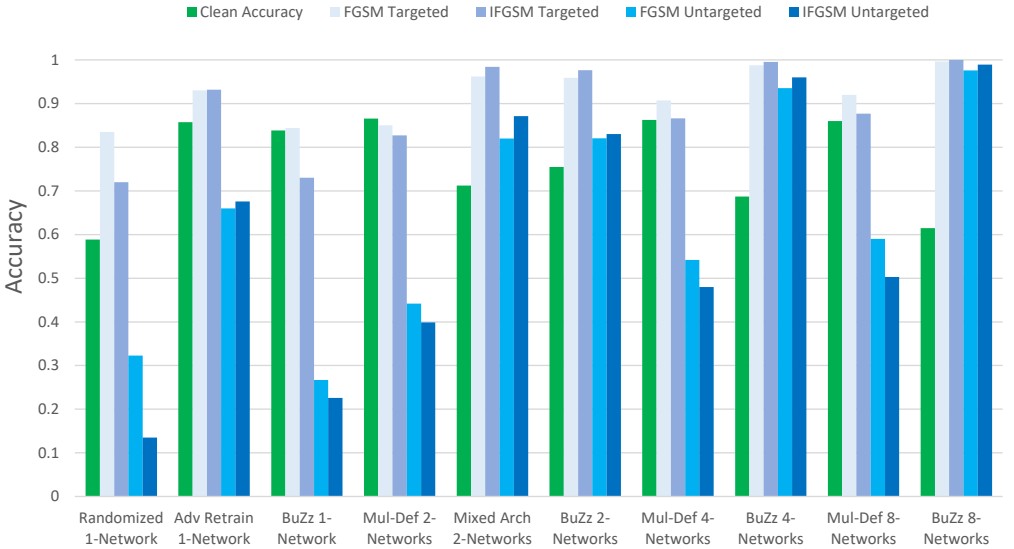

Figure 3: Mixed black-box attack on defenses Xie et al. (2018) (randomized 1 network), Guo et al. (2017) (mul-sec 1 network), and Srisakaokul et al. (2018) (mul-def 2, 4, and 8 network) for CIFAR-10. The setup of the attacks for this experiment is the same as the attacks in Section 4 and Supplemental Materials C and D.

(Xie et al. (2018)) has a single network and uniformly selects an image transformation from an a-priori fixed set of a small number of image transformations to defeat white-box attacks. In the white-box setting (Athalye et al. (2018a)) shows that this defense does not work. But is this defense secure against black-box attacks? To maintain a sufficiently high clean accuracy, the random image transformations should not have high randomness. Hence, the boundaries of any single network/classifier and the network/classifier with one of the random image transformations may be highly aligned. This implies that the adversarial examples created for any classifier will likely transfer to the network with randomization operations. This is confirmed by experiments reported in the column "Randomized 1 Network" in Figure 3. We can see that the defense accuracy (i.e., clean accuracy for the defense model) drops to 0.59 and the attacker's success rate of the untargeted mixed black-box attack (using iFGSM) is 0.865 (the table entry depicts the defense rate 0.135 defined as 1 minus the attacker's success rate).

Similarly, the defense proposed in (Meng & Chen (2017)) – a defense with a single network and multiple different auto-encoders as image transformations from which one is selected at random per query – is not secure against pure black-box attacks, i.e., the success rate of targeted pure black-box attack on the defense using C&W attack for CIFAR-10 and MNIST is at least 99%, see (Carlini & Wagner (2017b)).

In (Guo et al. (2017)), the designer selects a set of possible image transformations for a single network and keeps the selection of the chosen image transformation secret. The image transformation will distort the noise as explained in (Guo et al. (2017)). This is BUZz for a single protection layer (without multiple networks and threshold voting). However, there is no buffer zones for any single network and thus, there exist many adversarial examples with small noises. We have validated this claim for BUZz with a single protection layer because it is very close to the one in (Guo et al. (2017)). In column "BUZz 1-Net" in Figure 3, the best attack is untargeted one and the success rate for CIFAR-10 is 77% (corresponding to defense rate 0.226).

**Black-Box Defenses based on Multiple Networks.** In (Liu et al. (2017)), the authors study the transferability between different networks which have different structures for the ImageNet dataset. The authors report that the transferability between the networks is small (claimed to be 'close to zero'). For this reason, it may be possible to have the BUZz defense where protected layers represent different networks with different architectures. However, experimentally we have built a defense with VGG16 and Resent20 trained on CIFAR10. As reported in Table 4 and Figure 2, the 2-net

BUZz with 2 VGG16 and 2-net BUZz with 1 VGG and 1 Resnet (Mixed Arch 2-Net) have the same performance. It implies that having different architecture does not give us any advantage in BUZz defense.

In unpublished work (Srisakaokul et al. (2018)) the authors have proposed a defense against white-box attacks based on multiple networks with the *same* architecture. The authors develop their defense based on a retraining technique. First, the authors apply adversarial attacks on each network to generate a set of adversarial examples. For example, for each network $j$ a white-box attack produces a set of adversarial examples $\mathcal{S}_j$. Next network $j$ will be retrained with the clean training data set together with some of the adversarial sets $\mathcal{S}_h$, $h \neq j$. The authors argue that all the networks cannot be fooled at the same time for a given adversarial example and this leads to a low(er) attacker's success rate. The final outputted class label is the predicted label of one of the networks chosen at random among all networks; this gives high clean accuracy.

With respect to white-box attacks, the defense in (Srisakaokul et al. (2018)) seems not secure: For verifying resistance against white-box attacks, (Srisakaokul et al. (2018)), only attacks each model separately instead of attacking all the models at the same time as is done in Ensemble Pattern Attacks in Section 4 in (Xie et al. (2018)). Hence, the authors should do the Ensemble Pattern Attack on their defense to have a completed claim on white-box resistance.

For testing resistance against Papernot's black-box attack, the authors only work with an initial set of 150 samples and 5 runs. This gives an augmented set of only $2^4 \cdot 150 = 2400$ samples used for building the synthetic model (compared to an augmented set of $2^5 \cdot 50\text{K}$ samples in our experiments). Hence, the performance of synthetic model $g$ is very poor and as a result a lower attacker's success rate. Nevertheless, even with a poor synthetic network (that is, very weak black-box adversary) the reported success rate of the Papernot's attack with FGSM is still high, i.e., around $18\%/27\%$ for MNIST and CIFAR-10 (see Table 5 in (Srisakaokul et al. (2018))). We performed experiments for our strong mixed black-box adversary and found an attacker's success rate of $50\%$ for the best attack on the best defense (Mul-Def 8 Network in Figure 3).

We summarize the attacker's success rate of the black-box attacks on the aforementioned defenses in Table 5. In the table the *original accuracy* denotes the clean accuracy for the classifier without implemented defense and *defender accuracy* denotes the clean accuracy with implemented defense. We can see that all defenses do not want to sacrifice clean accuracy $p$ – generally at most a drop of about 5%, see (Meng & Chen (2017)), with a larger drop of 25% in (Xie et al. (2018)). As a result of aiming at keeping the clean accuracy the same, the attacker's success rate $\alpha$ remains very high, typically $\alpha \geq 0.75$ for experiments with *MNIST, CIFAR-10, and CIFAR-100 as in this paper* with (Tramèr et al. (2017); Srisakaokul et al. (2018)) being the exception with about $50\%$. These high attacker's success rates give a drop in the effective clean accuracy of $\delta = p - (p - \gamma)(1 - \alpha) \geq p - p(1 - 0.5) = 0.5p$; the $\delta$ values are listed in the table. We do not differentiate the $\delta$ values for targeted resp. untargeted attacks. We see that $\delta \sim 0.29, 0.43$ for (Tramèr et al. (2017); Srisakaokul et al. (2018)) and $\delta \geq 0.65$ for other defenses.

For ImageNet, not studied in our paper, we have $\delta$ equal to 0.211 in (Tramèr et al. (2017)) and 0.396 in (Kurakin et al. (2016)), albeit for the weaker pure black-box adversary (without access to the entire training data set). First, for the stronger adversary analysed in this paper we expect these $\delta$ values to become higher. Second, since Tramèr et al. (2017) has (compared to $\delta = 0.247$ for BUZz) a higher $\delta = 0.29$ for the stronger adversary for CIFAR-10, we expect to also see this reflected in a smaller $\delta$ for ImageNet for BUZz compared to Tramèr et al. (2017) (after fine tuning image transformations). We leave exact experiments for future work. Third, even if BUZz would do similar when comparing $\delta$ to Tramèr et al. (2017) for ImageNet, BUZz would still be the better choice: This is because even though the $\delta$ values are the same, the attack success rate for BUZz is expected to be 5-10%, much less than $27\%$ of Tramèr et al. (2017) for the weaker adversary. This means that there is much less adversarial control in BUZz compared to Tramèr et al. (2017). (This has already been shown to be true for CIFAR-10 in Table 5.)

## C    PSEUDO ALGORITHMS: BLACK-BOX ATTACK & BUZZ

**Synthetic network.** Algorithm 1 depicts the construction of a synthetic network $g$ for the oracle based black-box attack from Papernot et al. (2017). The attacker uses as input an oracle $\mathcal{O}$ which

| Defense | Data set | Attack | Att. success rate | Or. Accuracy | Def. Accuracy | $\delta$ |
|---|---|---|---|---|---|---|
| (Tramèr et al. (2017)) | ImageNet | Pure BB -FGSM | $\approx 27\%$ (Tramèr et al. (2017)) | 78% (ImageNet) | 78% (ImageNet) | 0.211 |
| (Kurakin et al. (2016)) | ImageNet | Pure BB- FGSM | $\geq 50\%$ (Kurakin et al. (2016)) | 78.4%(ImageNet) | 77.6% (ImageNet) | 0.396 |
| (Tramèr et al. (2017)) | CIFAR-10 | Mixed BB -FGSM | 34% (this paper) | 85% (CIFAR-10) | 85% (CIFAR-10) | 0.29 |
| (Srisakaokul et al. (2018)) | CIFAR-10 | Mixed BB - iFGSM | 50% (this paper) | 86% (CIFAR-10) | 86% (CIFAR-10) | 0.43 |
| (Guo et al. (2017)) | CIFAR-10 | Mixed BB - iFGSM | 77% (this paper) | 84% (CIFAR-10) | 84% (CIFAR-10) | 0.65 |
| (Feinman et al. (2017)) | CIFAR-10 | Pure BB - C&W | $\geq 80\%$ (Carlini & Wagner (2017a)) | 82.6% (CIFAR-10) | 82.6% (CIFAR-10) | 0.661 |
| (Xie et al. (2018)) | CIFAR-10 | Mixed BB - iFGSM | 86.5% (this paper) | 84% (CIFAR-10) | 59% (CIFAR-10) | 0.76 |
| (Meng & Chen (2017)) | MNIST & CIFAR-10 | Pure BB - C&W | $\geq 99\%$ (Carlini & Wagner (2017b)) | 90.6% (CIFAR-10) | 86.8% (CIFAR-10) | 0.897 |
| (Papernot et al. (2016a)) | MNIST | Oracle BB - FGSM | 70% (Papernot et al. (2017)) | 99.51% (MNIST) | 98.14% (MNIST) | 0.701 |
| (Metzen et al. (2017)) | MNIST | Pure BB - C&W | $\geq 84\%$ (Carlini & Wagner (2017b)) | 91.5% (MNIST) | 91.5% (MNIST) | 0.769 |

Table 5: Attacker's success rate of black-box attacks for state-of-the-art defenses

represents black-box access to the target model $f$ which only returns the final class label $F(f(x))$ for a query $x$ (and not the score vector $f(x)$). Initially, the attacker has (part of) the training data set $\mathcal{X}$, i.e., he knows $\mathcal{D} = \{(x, F(f(x))) : x \in \mathcal{X}_0\}$ for some $\mathcal{X}_0 \subseteq \mathcal{X}$. Notice that for a single iteration $N = 1$, Algorithm 1 therefore reduces to an algorithm which does not need any oracle access to $\mathcal{O}$; this reduced algorithm is the one used in the pure black-box attack (Carlini & Wagner (2017b); Athalye et al. (2018a); Liu et al. (2017)). In this paper we assume the strongest black-box adversary in Algorithm 1 with access to the entire training data set $\mathcal{X}_0 = \mathcal{X}$ (notice that this excludes test data for evaluating the attack success rate).

In order to construct a synthetic network the attacker chooses a-priori a substitute architecture $G$ for which the synthetic model parameters $\theta_g$ need to be trained. The attacker uses known image-label pairs in $\mathcal{D}$ to train $\theta_g$ using a training method $M$ (e.g., Adam (Kingma & Ba (2014))). In each iteration the known data is doubled using the following data augmentation technique: For each image $x$ in the current data set $\mathcal{D}$, black-box access to the target model gives label $l = \mathcal{O}(x)$. The Jacobian of the synthetic network score vector $g$ with respect to its parameters $\theta_g$ is evaluated/computed for image $x$. The signs of the column in the Jacobian matrix that correspond to class label $l$ are multiplied with a (small) constant $\lambda$ – this constitutes a vector which is added to $x$. This gives one new image for each $x$ and this leads to a doubling of $\mathcal{D}$. After $N$ iterations the algorithm outputs the trained parameters $\theta_g$ for the final augmented data set $\mathcal{D}$.

---

**Algorithm 1** Construction of synthetic network $g$ in Papernot's oracle based black-box attack

---

1: **Input:**
2:      $\mathcal{O}$ represents black-box access to $F(f(\cdot))$ for target model $f$ with output function $F$;
3:      $\mathcal{X}_0 \subseteq \mathcal{X}$, where $\mathcal{X}$ is the training data set of target model $f$;
4:      substitute architecture $G$
5:      training method M;
6:      constant $\lambda$;
7:      number $N$ of synthetic training epochs
8: **Output:**
9:      synthetic model $g$ defined by parameters $\theta_g$
10:      ($g$ also has output function $F$ which selects the max confidence score;
11:      $g$ fits architecture $G$)
12:
13: **Algorithm:**
14: **for** $N$ iterations **do**
15:      $\mathcal{D} \leftarrow \{(x, \mathcal{O}(x)) : x \in \mathcal{X}_t\}$
16:      $\theta_g = M(G, \mathcal{D})$
17:      $\mathcal{X}_{t+1} \leftarrow \{x + \lambda \cdot \mathrm{sgn}(J_{\theta_g}(x)[\mathcal{O}(x)]) : x \in \mathcal{X}_t\} \cup \mathcal{X}_t$
18: **end for**
19: Output $\theta_g$

---

The precise set-up for our experiments is given in Tables 6, 7, and 8. Table 6 details the used training method $M$ in Algorithm 1. For the evaluated data sets Fashion MNIST, CIFAR-10, and CIFAR-100 without data augmentation, we enumerate in Table 7 the amount $|\mathcal{X}_0|$ of training data together with parameters $\lambda$ and $N$ in Algorithm 1 ($\lambda = 0.1$ and $N = 6$ are taken from the oracle based black-box attack paper of Papernot et al. (2017); notice that a test data set of size 10.000 is standard practice; all remaining data serves training and this is *entirely* accessible by the attacker).

Table 8 depicts the architecture $G$ of the CNN network of the synthetic network $g$ for the different data sets; the structure has several layers (not to be confused with 'protection layer' in BUZz which is an image transformation together with a whole CNN in itself). The adversary attempts to attack BUZz and will first learn a synthetic network $g$ with architecture $G$ (used as input in Algorithm 1) that corresponds to Table 8. Notice that the image transformations are kept secret and for this reason the attacker can at best train a synthetic vanilla network. Of course the attacker does know the set from which the image transformations in BUZz are taken and can potentially try to learn a synthetic CNN for each possible image transformation and do some majority vote (like BUZz) on the outputted labels generated by these CNNs. However, there are exponentially many transformations making such an attack infeasible. For future research we will investigate whether a small sized subset of 'representative' image transformations can be used to generate a synthetic model which can be used to attack BUZz in a more effective way. Nevertheless, we believe that BUZz will remain secure because of the security argument given in Section 3 where is shown how a single perturbation $\eta$ leads to very different perturbations at each protected layer in BUZz. This leads to 'wide' buffer zones and their mere existence is enough to achieve our security goal – security is not derived from keeping the image transformations private. Keeping these transformations private just makes it harder for the adversary to construct a more effective attack but the resulting attack is expected to still have small attacker's success rates. We leave this study for future work.

Table 6: Training parameters used in the experiments

| Training Parameter | Value |
|---|---|
| Optimization Method | ADAM |
| Learning Rate | 0.0001 |
| Batch Size | 64 |
| Epochs | 100 |
| Data Augmentation | None |

Table 7: Papernot Black-Box Attack Parameters

| | $|\mathcal{X}_0|$ | $N$ | $\lambda$ | Testing set |
|---|---|---|---|---|
| CIFAR-10 | 50000 | 6 | 0.1 | 10000 |
| CIFAR-100 | 50000 | 6 | 0.1 | 10000 |
| Fashion MNIST | 60000 | 6 | 0.1 | 10000 |

Table 8: Architectures of synthetic neural networks $g$ from (Carlini & Wagner (2017a))

| Layer Type | Fashion MNIST and CIFAR-10 | CIFAR-100 |
|---|---|---|
| Convolution + ReLU | $3 \times 3 \times 64$ | $3 \times 3 \times 64$ |
| Convolution + ReLU | $3 \times 3 \times 64$ | $3 \times 3 \times 64$ |
| Max Pooling | $2 \times 2$ | $2 \times 2$ |
| Convolution + ReLU | $3 \times 3 \times 128$ | $3 \times 3 \times 128$ |
| Convolution + ReLU | $3 \times 3 \times 128$ | $3 \times 3 \times 128$ |
| Max Pooling | $2 \times 2$ | $2 \times 2$ |
| Fully Connected + ReLU | 256 | 256 |
| Fully Connected + ReLU | 256 | 256 |
| Softmax | 10 | 100 |

**White-box attack on the synthetic network.** The targeted iterative Fast Gradient Sign Method (FGSM) of (Goodfellow et al. (2014)) is given in Algorithm 2 (for our implementation we use the cleverhans library, see `https://github.com/tensorflow/cleverhans`). The non-iterative variant has outer loop size $H = 1$. For a untargeted attack no adversarial label $l'$ is given as input and the loss function $L$ is only a function of $x$, $l$, and $\theta_g$ (the loss function $L$ should be properly defined to make the attack targeted attack or untargeted). The algorithm walks in $H$ iterations along

the gradient towards the boundary of $x$'s region (where a region with label $l'$ should start). The maximum perturbation will have entries in the interval $[-\epsilon, +\epsilon]$ (since each of the $H$ iterations at most add $\epsilon/H$).

When Algorithm 2 is used as a pure black-box attack, then no oracle access is available and comparison $l' = \mathcal{O}(x)$ is replaced by comparison $l' = F(g(x))$, which uses the synthetic network. In this paper we assume the stronger black-box adversary who has oracle access in Algorithm 2.

---

**Algorithm 2** Targeted iterative Fast Gradient Sign Method

---

 1: **Input:**
 2:   $\mathcal{O}$ represents black-box access to $F(f(\cdot))$ for target model $f$ with output function $F$;
 3:   parameters $\theta_g$ of a synthetic network $g$;
 4:   loss function $L$ for the attack,
 5:   threshold $\epsilon > 0$,
 6:   outer iteration size $H$,
 7:   benign image $x$ with true label $l$,
 8:   adversarial target label $l'$,
 9: **Output:**
10:   adversarial example $x'$
11:
12: **Algorithm:**
13: **for** $H$ iterations **do**
14:   $x = x - (\epsilon/H) \cdot \mathrm{sgn}(\nabla_x L(x, l, l'; \theta_g))$
15:   **if** $l' = \mathcal{O}(x)$ **then**
16:    output $x' = x$
17:    break;
18:   **end if**
19: **end for**

---

In (Carlini & Wagner (2016)), the authors propose a powerful non-iterative attack (coined as C&W attack), which can also be used to produce adversarial examples with small noise. The authors use some optimization method OPT (e.g Adam (Kingma & Ba (2014))) to find $\eta$ which minimizes $\|\eta\|_p + c \cdot L(x + \eta, l, \theta_g)$ for a untargeted attack (or $\|\eta\|_p - L(x + \eta, l, l', \theta_g)$ for a targeted attack); the output of OPT is $x' = x + \eta$. Here, $\|\cdot\|_p$ denotes the $p$-norm, $L$ is the objective function defined in Carlini & Wagner (2016) for a given synthetic network $g$ with parameters $\theta_g$. We do not use the C&W algorithm of (Carlini & Wagner (2016)) because it achieves a lower attacker's success rate compared to iterative FGSM in our setup: For example, the C&W[7] L2 untargeted attacker's success rate for 1000 samples is only 0.006 on our 2 Network BUZz CIFAR-10 defense. This translates to an overwhelmingly large 99.4% defense rate. Contrast this to the attacker's success rate based on the untargeted iFGSM for our 2-Network BUZz defense which is $\approx 0.186 \gg 0.006$ (see Table 2).

Table 9: Parameters for iFGSM and FGSM in the mixed black-box attack on BUZz for CIFAR-10, CIFAR-100 and Fashion MNIST

|  | $\epsilon_{untargeted}$ | $H_{untargeted}$ | $\epsilon_{targeted}$ | $H_{targeted}$ |
|---|---|---|---|---|
| CIFAR-10 | 10/256 | 10 | 1/20 | 10 |
| CIFAR-100 | 10/256 | 10 | 1/20 | 10 |
| Fashion MNIST | 0.1 | 10 | 0.3 | 10 |

We enumerate in Table 9 the parameters $\epsilon$ and $H$ used in our experiments – we have taken these from literature (Meng & Chen (2017)). Notice that literature often reports $\epsilon/H$ as the step size in FGSM while we list values for $\epsilon$ which corresponds to the size of the final perturbation that leads to the adversarial example (and as such we can interpret $\epsilon$ as the threshold for noise being visually perceptible by the human eye).

---

[7]We use the library in `https://github.com/carlini/nn_robust_attacks` to implement the C&W attack.

**Success rate black-box attack.** In order to implement the black-box attack we first run Algorithm 1 which outputs the parameters of a synthetic network $g$. Next, out of the test data (each data set has 10.000 samples in our set-up) we randomly select 1000 (this is custom in literature) samples $(x, l)$ which the target model $f$ (i.e., BUZz in this paper) correctly classifies. For each of the 1000 samples we run Algorithm 2 to produce 1000 adversarial examples. The attacker's success rate is the fraction of adversarial examples which change $l$ to the desired new randomly selected $l'$ in a targeted attack or any other label $l'$ for an untargeted attack.

**Image transformations for BUZz.** In the BUZz, we use image transformations that are composed of a resizing operation $i(x)$ and a linear transformation $c(x) = Ax + b$. An input image $x$ at a protected layer in BUZz is linearly transformed into an image $i(c(x))$ before it enters the corresponding CNN network with VGG16 architecture for CIFAR-10 and CIFAR-100 (see `https://neurohive.io/en/popular-networks/vgg16/`8) or minivgg architecture for FashionMNIST (see `TODO`). In a network implementation one can think of $i(c(x))$ as an extra layer in the CNN architecture of VGG16 itself (here 'layer' should not be confused with the terminology 'protected layer').

For the resize operations $i(\cdot)$ used in each of the protected layers in BUZz, we choose sizes that are larger than the original dimensions of the image data. We do this to prevent loss of information in the images that down sizing would create (and this would hurt the clean accuracy of BUZz). In our experiments we use BUZz with 2, 4, and 8 protected layers. Each protected layer gets its own resize operation $i(\cdot)$. When using 8 protected layers, we use image resizing operations from 32 to 32, 40, 48, 64, 72, 80, 96, 104. Each protected layer will be differentiated from each other protected layer due to the difference in how much resizing each layer implements. This will lead to less transferability between the protected layers and as a result we expect to see a wider buffer zone which diminishes the attacker's success rate. When using 4 protected layers, we use a copy of the 4 protected layers from BUZz with 8 networks that correspond to the image resizing operations from 32 to 32, 48, 72, 96. When using 2 protected layers, we use a copy of the 2 protected layers from BUZz with 8 networks that correspond to the image resizing operations from 32 to 32 and 104. In our implementation we use resizing operation from github `https://github.com/cihangxie/NIPS2017_adv_challenge_defense` (Xie et al. (2018)).

For each protected layer, the linear transformation $c(x) = Ax + b$ is randomly chosen from some statistical distribution (the distribution is public knowledge and therefore known by the adversary). Design of the statistical distribution depends on the complexity of the considered data set (in our case we experiment with FashionMNIST, CIFAR-10, and CIFAR-100). Transformation $c(x)$ takes an image of size $n_1 \times n_2 \times_3$ as input and considers this as a vector of length $k = n_1 n_2 n_3$. Here, $n_1$ and $n_2$ denote the horizontal and vertical width in pixels of image $x$; $n_3 = 3$ means that each pixel has a red, blue, and green values; $n_3 = 1$ means that each pixel only has one black/white value. CIFAR-10 and CIFAR-100 have $32 \times 32 \times 3$ images and FashionMNIST has $28 \times 28 \times 1$ images. All the values in vector $x$ are converted from integers $[0..255]$ to the range $[-0.5, +0.5]$ of real numbers. Notice that the entries of $c(x)$ may have their values outside of this range.

In our implementation we do not consider $x$ to be in vector representation; we think of $x$ as $n_3$ times a $n_1 \times n_2$ matrix. For example, $x = (X_1, X_2, X_3)$ for $n_3 = 1$. We restrict $c(x) = Ax + b$ to linear operations

$$c(X_1, X_2, X_3) = (X_1 A_1 + b_1, X_2 A_2 + b_2, X_3 A_3 + b_3),$$

where $A_i$ are $n_2 \times n_2$ matrices and $b_i$ are $n_1 \times n_2$ matrices.

For CIFAR-10 and CIFAR-100 we take matrices $A_i$ to be identity matrices (this also makes $A$ the identity matrix in the vector representation of $c(x)$) and we use the same matrix $b$ for each of the matrices $b_i$, i.e.,

$$b' = b_1 = b_2 = b_3.$$

This means that we use the same random offset in the red, blue, and green values of a pixel. The reason for making this design decision is because for CIFAR-10 and CIFAR-100 we found that fully random $A$ creates large drops in clean accuracy, even when the network is trained to learn such distortions. As a result, for data sets with high spatial complexity like CIFAR-10 and CIFAR-100, we do not select $A$ randomly. We choose $A$ to be the identity matrix. Likewise for $b'$ we only randomly generate 35% of the matrix values and leave the rest as 0. For the randomly generated values, we choose them from a uniform distribution from $-0.5$ to $0.5$.

For datasets with less spatial complexity like FashionMNIST, we equate matrices $A' = A_1 = A_2 = A_3$ and $b' = b_1 = b_2 = b_3$ and select $A'$ and $b'$ as random matrices: The values of $A'$ and $b'$ are selected from a Gaussian distribution with $\mu = 0$ and $\sigma = 0.1$.

## D EXPERIMENTAL RESULTS

We reported our experimental results for CIFAR-10 in Section 4 for 5 runs (see Table 2). Here, one run is implemented by first choosing matrices $A'$ and $B'$ from the distribution corresponding to the considered data set for each protected layer. Next the attacker's success rate and clean accuracy of the defense are simulated. For each next run, matrices $A'$ and $B'$ are again chosen anew.

As shown in Figure 5 for CIFAR-10, the average result for 5 runs is not much different from that of 1 run. Moreover, we also report the result of the best case of the attack among 5 runs in Table 10. As one can see, different runs give very similar results. This shows that BUZz is not sensitive to the choice of $A'$ and $B'$ (worst and best cases are close to one another).

Table 10: Mixed black-box attack on CIFAR-10– the best case

|                  | Vanilla | 2-Networks | 4-Networks | 8-Networks |
|------------------|---------|------------|------------|------------|
| Clean Accuracy   | 0.8835  | 0.76       | 0.69       | 0.62       |
| FGSM Targeted    | 0.831   | 0.96       | 0.99       | 0.99       |
| iFGSM Targeted   | 0.766   | 0.98       | 0.99       | 1          |
| FGSM Untargeted  | 0.297   | 0.85       | 0.94       | 0.97       |
| iFGSM Untargeted | 0.282   | 0.87       | 0.97       | 0.99       |

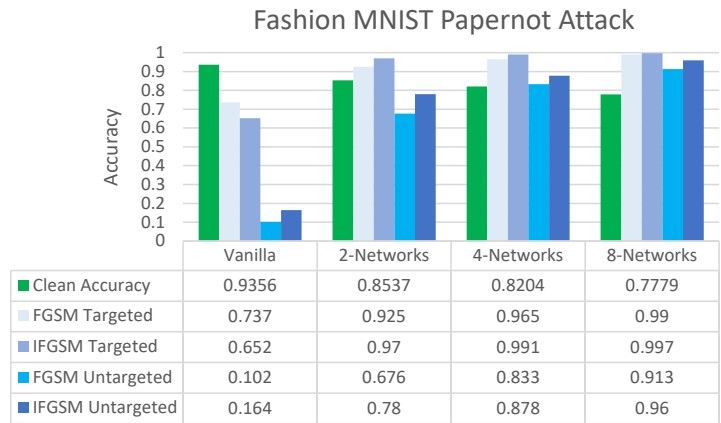

Figure 4: Mixed Black-Box Attack on FashionMNIST, single run.

For completeness, we also did experiments for pure black-box attacks. The results are in Figures 7, 8, and 9 for a single run. The numbers confirm the mixed black-box attack being stronger than the pure black-box attack.

We also mention that for the vanilla network for CIFAR-10 `https://github.com/kuangliu/pytorch-cifar` shows a clean accuracy of 92.6% which is more than the reported 88.4% in our experiments – this is due to our limited training time (limited resources). For the vanilla network for CIFAR-100 without data augmentation `https://github.com/SamKirkiles/vgg-cifar100` shows a clean accuracy of 64% which is similar to the clean accuracy of 63% reported here. For the vanilla network for FashionMNIST `https://www.pyimagesearch.com/2019/02/11/fashion-mnist-with-keras-and-deep-learning/` shows a clean accuracy of 94% which is equal to the clean accuracy reported here.

**Discussion.** From the experiments on FashionMNIST, CIFAR-10, CIFAR-100, we see the success of untargeted mixed black-box attacks on vanilla nets is significantly higher than that of the targeted

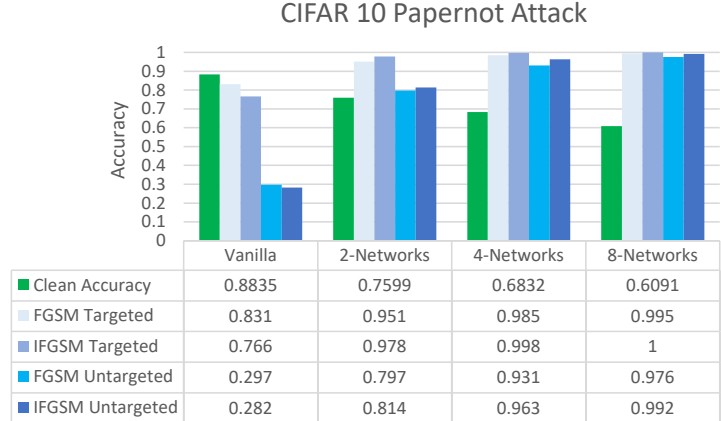

Figure 5: Mixed Black-Box Attack on CIFAR-10, single run.

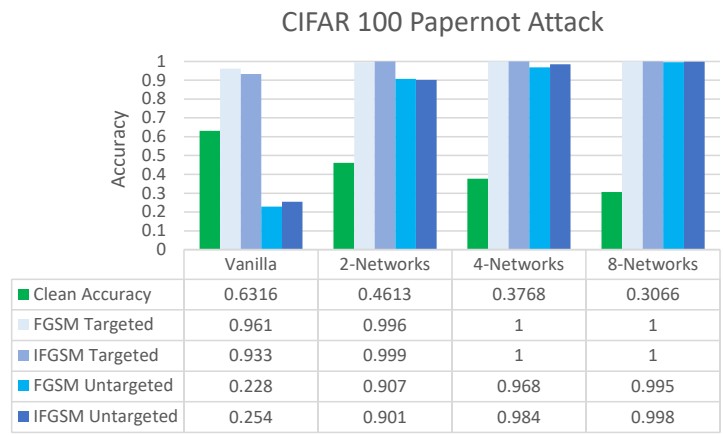

Figure 6: Mixed Black-Box Attack on CIFAR-100, single run.

attacks. Moreover, the success of targeted attacks on the vanilla network of CIFAR-100 is close to $1\%$ which is significantly poorer than that of CIFAR-10 or FashionMNIST (i.e., $16\% - 19\%$). We believe the main reason is that the clean accuracy of the vanilla network of CIFAR-100 is significantly lower than that of CIFAR-10 or FashionMNIST, i.e., $63\%, 88\%$ and $93.5\%$, respectively. From this result, we also believe that the resistance of a network or classifier against adversarial machine learning strongly depends on the clean accuracy of the network itself; the lower the clean accuracy, the lower the attacker's success rate as well (remember that the attacker's success rate is only measured over images that are accurately predicted by the network).

From the view of the designer, she/he should pay attention to the highest threat or the most powerful attack, in this case untargeted mixed black box attacks. In other words, we should use $\delta_u$ as a true measurement to evaluate the resistance of a given defense. Equipped with this argument, we can now see that BUZz has very good resistance against adversarial examples at the cost of clean accuracy. As we argued in Table 5, current-state-of-the-art defenses generally have very large $\delta_u$ which is due to poor resistance against adversarial examples – this is because their clean accuracy is nearly equal to that of the vanilla network (the defenses do not want to give up some of the clean accuracy). A good lesson we can learn from Table 3 is that we have to sacrifice something (here, clean accuracy) to gain security. If the defense does not have 'buffer zones', then the adversary always wins the game in that it is possible to produce with significant probability adversarial examples with small noise to bypass the defense. Including 'buffer zones' means the designer has to give up some clean accuracy.

For Fashion MNIST, we may want to use 8-network BUZz because it has the best trade-off between defense rate (one minus the attacker's succes rate) and the defense accuracy (the clean accuracy of the defense). Similarly, for CIFAR-10 and CIFAR-100, we suggest 4-network BUZz and 2-network BUZz, respectively.

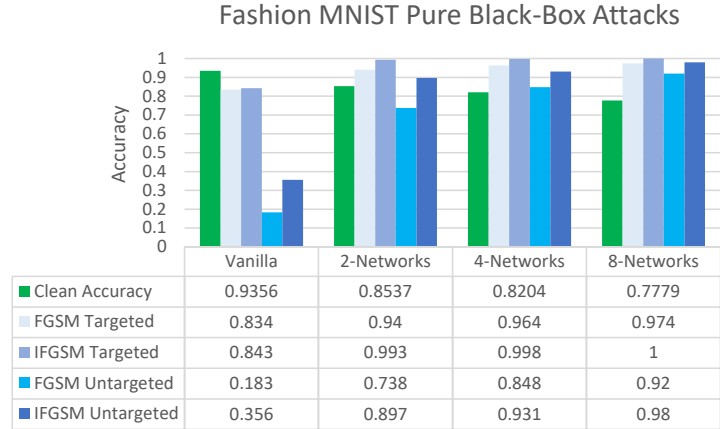

Figure 7: Pure Black-box Attack on FashionMNIST, single run.

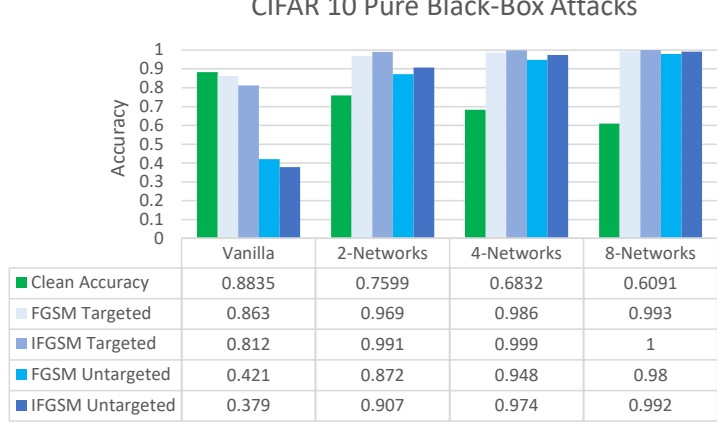

Figure 8: Pure Black-box Attack on CIFAR-10, single run.

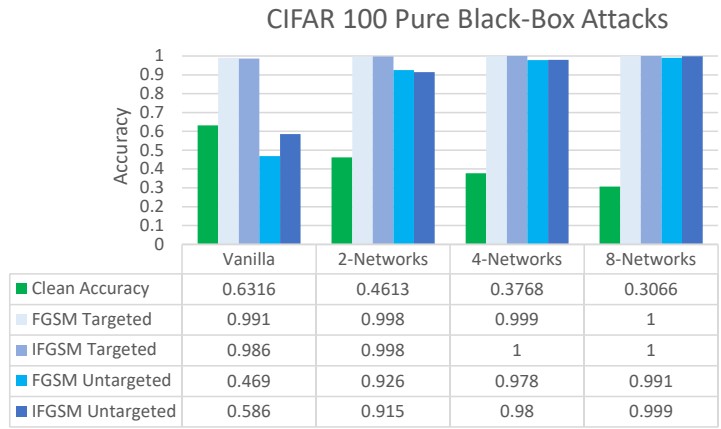

Figure 9: Pure Black-box Attack on CIFAR-100, single run.

