# OpenReview forum: "BUZz: BUffer Zones for defending  adversarial examples in image classification"
_ICLR.cc/2020/Conference — Reject_

### Official Review · AnonReviewer2 · 2019-10-18
**Official Blind Review #2**

**Rating:** 1

**Review:**

This paper proposes a defense against black box adversarial attack. The authors train an ensemble of deep networks, and output a null label when the ensemble disagree. Success of adversarial attack is defined as the ensemble outputs an incorrect label that is not null. The authors experimentally show improved robustness to adversarial attack.

The idea is itself new, but very similar ideas are well known in the literature, and it is difficult to conclude that the proposed approach is superior. Several examples are:

Defense by majority vote with ensembles has appeared several times in the literature (e.g. Pang et al 2019). The pro is that this paper proposes a novel way to create an ensemble by applying random linear transformations and rescaling to the input. But it is not clear this is superior compared to existing methods.

Randomized smoothing (Cohen et al, 2019) guarantees smoothness of the classifier (and thus robustness to perturbation attack under certain norms). Note that randomized smoothing  provides certified guarantee against a stronger attack model (white box); it also guarantees the size of the margin (or buffer as the authors call it). The intuition of this paper is based on similar ideas so it seems necessary to at least compare with randomized smoothing.

Outputting a null label is a major workhorse of adversarial defense. For example, previous work use a generative model to detect out of distribution samples; use a calibrated classifier to output null when low confidence.

Because the idea is only mildly interesting, good experimental support becomes crucial. However, I think there are several short-comings with the experiments:

The experiment contains only one (fairly old) attack method. Several recent alternatives such as SimBA (Guo et al 2019) can make the experiments more convincing.

The architecture is no longer the same for the target model (which is an ensemble with an additional random transformations) and surrogate model. It is unclear if the improvement is simply because of the difference in architecture.

The comparison to baselines seem unfair because it seems that the compared baselines do not have the option of outputting the null label. For example, a simple baseline of randomized smoothing + output the null label if the logit scores are below a threshold can make the story much stronger.

Minor comments:

Several suggestions on writing: the introduction contains much technical detail and even experimental results, and these are repeated again in later sections. The experimental section has many minor implementation details that could go into the appendix.


**Experience Assessment:**

I have published one or two papers in this area.

**Review Assessment: Checking Correctness Of Derivations And Theory:**

I assessed the sensibility of the derivations and theory.

**Review Assessment: Checking Correctness Of Experiments:**

I assessed the sensibility of the experiments.

**Review Assessment: Thoroughness In Paper Reading:**

I read the paper at least twice and used my best judgement in assessing the paper.

---

> ### Author Response · Authors · 2019-11-14
> **Response to Review 2**
>
> "The idea is itself new, but very similar ideas are well known in the literature, and it is difficult to conclude that the proposed approach...”
>
> Answer: Indeed, we are not aware of this paper. In Peng et al. 2019, the authors proposed a defense based on several classifiers. The good point of this defense is that it has high clean accuracy and has high robustness as shown in the paper. However, there are several components of the paper which are not completely clear. From these points it leads us to believe our defense is still a valid contribution. We list the points related to Peng et al. 2019 below:
>
> = White-box attacks: it is not clear to use that from the text that how the authors develop white-box attacks. We assume that the authors may blindly applied the attacks in section 2 to their defense. Actually, the defense may be vulnerable if we build the white-box attack as follows: given component classifiers F1(x), .., Fk(x), we want to build an objective function which optimizes the noise over F1(x),...,Fk(x) rather than F(x) = 1/k sum_{i=1}^k Fi(x). With this customized objective function for the attack, we may see that the efficiency of the defense may be reduced.
>
> = Black-box attacks: the authors did not perform Papernot's black box attack and a pure black attack. For comparison, we focus on CIFAR-10. First, they verified the transferability between their trained classifiers and from Figure 3 (untargeted attacks), the transferability between them is still very high, i.e., $60\%$. This implies that the success rate of pure black box attacks may be around $\alpha = 60\%$. Note that the noise is 0.05 which is much smaller in our case 0.1 for iterative FGSM. The considered black-box attack on our defense is much stronger, i.e., mixture black-box attack with 10-iteration FGSM with the noise 0.1.
>
> = In comparison to our work, their clean accuracy is very high in their defense $93.5%$ (Table 1) in this case, we can say that there is no drop in the clean accuracy, \gamma = 0. If we use our metric delta, then we will have delta = \gamma + (p-\gamma)*\alpha = 0 + (93.5\%-0)*60\%= 0.56. This means that this defense is not good against black-box attacks.
>
>  “Randomized smoothing (Cohen et al, 2019) guarantees smoothness ….”
>
> Answer: Indeed, we are not aware of this paper. We discuss an implementation of Cohen (Cao and Gang 2017) in Appendix B. Regarding these comments about different possible defenses, they bring up good points for comparison. However, these comments misunderstand the fundamental goal of this paper. This paper is designed to provide the first truly secure black-box defense, not evaluate other defenses which did not fully experiment with black-box attacks in their defense. While other techniques may exist, many of them remain untested. e.g. the smoothness defense suggested by the reviewer was never tested with Papernot’s black-box attack. The goal of our paper is not comparison but creation, the creation of a secure defense with rigorous experimental evidence to back it up.
>
> “Outputting a null label is a major workhorse of adversarial defense…”
>
> Answer: In terms of the diversity of attacks we consider, C&W, FGSM and IFGSM in conjunction with Papernot’s black-box attack. To address specific points related to the attacks mentioned by the reviewer: SimBA (Guo et al 2019) does not work with our defense because the attack requires the access to the Softmax layer of the classifier. However, this is prohibited in our defense. We did consider the C&W attack, but the performance of this attack is worse than iterative FGSM. Hence, we skipped this attack as mentioned in Section 4 - Experiment Section.
>
> “The architecture is no longer the same for the target model...”
>
> Ans: Architecturally diversity does not create a secure defense (it is necessary for our defense but not sufficient).
> This was shown in Papernot et al’s black-box attack paper where the synthetic model is different from the target model but the attack still works well. In addition we can see architectural diversity does not provide security by looking at different defense in figure 3 with very different architectures than the synthetic model.  The security reason for our defense is given in Section 3 (security argument).
>
> “The comparison to baselines seem unfair...”
>
>  Answer: According to our knowledge, there is no defense which produces null output because people do not want to sacrifice clean prediction accuracy for the security. The randomized smoothing with null label as you suggested is vulnerable as we explained in the related work (Appendix B - (Cao&Gong 2017)). If the x+\delta deeply locates in a different region, then the defense should fail. One important point for your suggested idea is that it is not clear to choose a threshold for that defense. Note that the original defense does not consider the null output or the usage of threshold because it may make the analysis more complicated for analysis.

---

### Official Review · AnonReviewer3 · 2019-10-21
**Official Blind Review #3**

**Rating:** 3

**Review:**

First of all, I think the authors do not do enough literature research on the topic of adversarial examples:

1. In Sec 2., the authors only mention the FGSM in White-box Attacks. It is widely accepted that when evaluating the white-box attacks, you should at least test PGD attacks and/or C&W attacks.

2. When you try to propose an adversarial defense, you should report the performance under white-box adaptive attacks. Only claiming effectiveness under black-box attacks is not informative or convincing.

The experiment results are also weird. For example, in Figure 8, why the Vanilla clean accuracy on CIFAR-10 is only 88.35%? Besides, the clean accuracy on CIFAR-10 of 2-Networks Buzz is 75%, 8-Networks Buzz is 60%. This clean performance is not acceptable, no matter how robust is the model.

**Experience Assessment:**

I have published in this field for several years.

**Review Assessment: Checking Correctness Of Derivations And Theory:**

N/A

**Review Assessment: Checking Correctness Of Experiments:**

I carefully checked the experiments.

**Review Assessment: Thoroughness In Paper Reading:**

I read the paper thoroughly.

---

> ### Author Response · Authors · 2019-11-14
> **Response to Review 3**
>
> Thank you for your comments. Here are our detailed responses.
>
> “First of all, I think the authors do not do enough literature research on the topic of adversarial examples:
> 1. In Sec 2., the authors only mention the FGSM in White-box Attacks. It is widely accepted that when evaluating the white-box attacks, you should at least test PGD attacks and/or C&W attacks.”
>
> Answer: Actually, we did test with C&W attacks on our defense in context of black-box attacks and it turns out that C&W is not as good as FGSM as we mentioned in Section 4 - Experimental Section. Moreover, FGSM allows us to precisely control the noise to observe the efficiency of the attack. Moreover, we also performed iterative FGSM in the attack which is conceptually not much different from PGD.
>
>  “2. When you try to propose an adversarial defense, you should report the performance under white-box adaptive attacks. Only claiming effectiveness under black-box attacks is not informative or convincing.”
>
>
> Answer: Modern machine learning security need to consider different adversarial settings. One extremely important area is machine learning services where companies like Google, Amazon, etc. train black-box classifiers for the customer, given some data. In this setting black-box attacks are the ONLY way to attack the classifier and hence securing these models is paramount. This type of setting is commonly accepted in the literature (see Papernot et al. ,” Practical Black-Box Attacks against Machine Learning”  (2016) which has been cited over 1000 times). The goal of this paper is NOT to consider white-box attacks, the aim of this paper is to create a defense against a black-box adversary.
>
>  “The experiment results are also weird. For example, in Figure 8, why the Vanilla clean accuracy on CIFAR-10 is only 88.35%? Besides, the clean accuracy on CIFAR-10 of 2-Networks Buzz is 75%, 8-Networks Buzz is 60%. This clean performance is not acceptable, no matter how robust is the model.”
>
> Answer: The objective of this paper is not to train a state-of-the-art classifier, the objective of this paper is to show what kind of penalty is incurred when a secure defense is implemented. Even if we use a classifier with above 90% accuracy for CIFAR-10, the accuracy will drop when using our defense. The important thing to note in the results is NOT the absolute accuracy, but the amount it changes by when implementing a defense (i.e. the difference between the vanilla accuracy and the accuracy in any of the defenses).
>
>  You mention that the clean performance is not acceptable for Buzz-8. We never suggest using an 8-network defense is acceptable or ideal. We merely provide this result to show the complete picture in terms of how clean accuracy can be traded for security. Before our paper, there was no defense that could guarantee security, period. Here we show that at the very least there is an option to get security (where before no secure option ever existed). For further proof of this please see Table 1 in our paper which uses the delta metric to mathematically provide proof of this claim.

---

### Official Review · AnonReviewer1 · 2019-10-25
**Official Blind Review #1**

**Rating:** 3

**Review:**

Summary: This paper proposes the concept of buffer zones and suggests to use unanimous voting as a way to induce such buffer zones. To widen the buffer zone, they further propose to diversity the model. They then proposed a new metric for measuring defense and demonstrated that their method is effective.

Decision: Weak Reject. This paper is fairly intuitive, but I am not sure about the fairness of the comparisons in the paper, and the level of rigor of the experiments.

I think the conjecture that buffer zones are widened when the models are diverse deserve to be empirically tested. It is not clear to me a prior how exactly the buffer zones widen (even though the belief that they widen is intuitively appealing). I think one way to potentially characterize the buffer zone is by actually performing white-box attack experiments comparing:
White-box attack vulnerability of unanimous voting vanilla models
White-box attack vulnerability of unanimous voting “diversified” models
I would also encourage the authors to think creatively about other ways to back up the the buffer zone claims put forth in the paper.

I am also a little puzzled with the authors’ choice of the diversification procedure. The procedure c(x) = Ax + b will only linearly transform each column of the image, but not each row. This design choice feels rather ad hoc. Why did the authors settle on Ax + b in particular? Why not xA + b? Or AxC + b?

Regarding the fairness of the experiments, do the models that the authors compare against have the luxury of returning a “Undecided” label? If not, then the problem formulation is fundamentally different, and I do not think the comparisons are necessarily fair. Are there any papers out there that also allow for an “Undecided” label? If so, they should be the baselines that one compares against. I have a rather hard time believing that this is the first paper to try unanimous voting across an ensemble.

I am generally inclined to switch to weak accept so long as the other reviewers are willing to accept that the experiments are sufficient and the comparisons are fair. I am not opposed to the new problem setting, since I think the setting makes sense. I just want to know that the paper is doing due diligence regarding related work in this setting.

**Experience Assessment:**

I do not know much about this area.

**Review Assessment: Checking Correctness Of Derivations And Theory:**

N/A

**Review Assessment: Checking Correctness Of Experiments:**

I assessed the sensibility of the experiments.

**Review Assessment: Thoroughness In Paper Reading:**

I read the paper at least twice and used my best judgement in assessing the paper.

---

> ### Author Response · Authors · 2019-11-14
> **Response to Review 1.**
>
> Thank you for your comments. Here are our detailed responses.
>
> This paper is fairly intuitive, but I am not sure about the fairness of the comparisons in the paper, and the level of rigor of the experiments.
>
> Ans: in our experiments, we implemented or studied many known defenses and black box attacks on them for  a fair and rigorous comparison (see Table 1) and appendix for related work and experiment.
>
> In experiment, we considered C&W attack and we found that it is not as good as FGSM. Hence, we skip the experiment for C&W.  We also explained why zeroth order black box attack (which is very similar to SimBA) does not work for our defense.   It means we did consider many common attacks for our defense.
>
>
> “I think the conjecture that buffer zones are widened when the models are diverse deserve to be empirically tested. It is not clear to me a prior how exactly the buffer zones widen (even though the belief that they widen is intuitively appealing). I think one way to potentially characterize the buffer zone is by actually performing white-box attack experiments comparing:
>
> White-box attack vulnerability of unanimous voting vanilla models
>
> White-box attack vulnerability of unanimous voting “diversified” models
>
> I would also encourage the authors to think creatively about other ways to back up the the buffer zone claims put forth in the paper.”
>
>
> Answer: While we agree with the author that more empirical evidence can be shown for our concept, the fact that increasing the number of networks (equivalent in some sense to increasing the size of the buffer zone) yields less successful adversarial samples that fool the defense (as shown in every experiment we conducted), strongly validates our buffer zone claim.
>
>
> “I am also a little puzzled with the authors’ choice of the diversification procedure. The procedure c(x) = Ax + b will only linearly transform each column of the image, but not each row. This design choice feels rather ad hoc. Why did the authors settle on Ax + b in particular? Why not xA + b? Or AxC + b?”
>
> Answer: This is done mainly for technical implementation reasons. We make our input X 1D (so that the A and b can be directly implemented as a linear non-learnable layer in a convolutional neural network). While other combinations are certainly possible, the goal of this paper is not to exhaustively explore every possible defense combination. Indeed, one could think of a multitude of different ways a transformation could be implemented within a network. For example, why do we put the transformation at the start instead of the middle of the network? Why do we use a linear transformation instead of applying non-linear activations in addition to the linear matrix multiplication? Why use one matrix A instead of multiple matrices? While these types of questions are all valid, they actually don’t truly consider the aim of the paper. The goal of this paper is to provide at least ONE possible concrete secure defense that has strong experimental evidence to back up its security claims. As long as we have one work defense framework (as given in this paper) the open problem of securing neural networks against black-box style attacks is solved. Other extensions (based on the working defense presented in this paper) can be considered as future work.
>
> “Regarding the fairness of the experiments, do the models that the authors compare against have the luxury of returning a “Undecided” label? If not, then the problem formulation is fundamentally different, and I do not think the comparisons are necessarily fair. Are there any papers out there that also allow for an “Undecided” label? If so, they should be the baselines that one compares against. I have a rather hard time believing that this is the first paper to try unanimous voting across an ensemble.”
>
>
> Answer: To the best of our knowledge, there is no defense producing "undecided" (or bottom) label as in our design. The reason is very simple: people want to maintain high clean accuracy rather than sacrifice it for security. We develop the metric \delta to show that many existing defenses with high clean prediction accuracy do not have any security as shown in Table 1.

---

> > ### Comment · AnonReviewer1 · 2019-11-15
> > **Response**
> >
> > Thanks for the response. Regarding my concern about fairness, I stand by my concern that the comparison with existing methods is, in a certain sense, fundamentally unfair since your model has the luxury of returning an "Undecided" label, whereas the models you test on do not have this luxury.
> >
> > That being said, this is not necessarily a bad thing. If it truly is the case that no existing defense employs the "Undecided" labeling convention, then it becomes a matter of whether allowing "Undecided" labeling is a worthwhile problem setting.
> >
> > At least in the context of this paper, the authors try to justify by saying the use of "Undecided" labeling allows for techniques that improve the proposed delta metric. However, it is not obvious to me that the delta metric is the correct quantity of interest. Ultimately, I believe it comes down to the cost of assigning an undecided versus a wrong label. The delta metric provides one such trade-off between undecided v wrong labels---though not a trade-off that I find intuitive. Without relying on the delta metric, I'm not comfortable that this paper in particular provides a convincing case for the use of "Undecided" labeling (this is not to say that a convincing case does not exist).
> >
> > If the other reviewers (and the AC) are convinced that this paper is the first to use "Undecided" labeling and furthermore provides a compelling case for the use of "Undecided" labels, then I think this paper has a compelling narrative:
> >
> > 1. Reframing the question to allow "Undecided" labels
> >
> > 2. Using unanimous voting to induce "Undecided" labels
> >
> > As of the moment, my assessment is that:
> >
> > 1. I'm not sure if this paper is the first to propose "Undecided" labeling. My suspicion is that work in the calibration community would have leveraged "Undecided" labeling as this point. I advise the authors to check the calibration literature and would also like to request the other reviewers to assess the novelty of "Undecided" labeling.
> >
> > 2. I do not think that this paper, as currently written, provides a compelling case for "Undecided" labeling.
> >
> > As such, my decision remains. I am happy to discuss further with the reviewers/AC if they wish to defend the authors on points #1 or #2.

---

> > > ### Author Response · Authors · 2019-11-15
> > > **Response to Review 1**
> > >
> > > Thank you for your helpful comments about the usage of "Undecided" class label.  Hopefully, our answer can somehow address your concern.
> > >
> > > 1.  We have checked the following calibration papers:
> > >
> > > 1. https://arxiv.org/abs/1702.01691
> > > 2. https://arxiv.org/pdf/1910.06259.pdf
> > > 3. https://openreview.net/pdf?id=Bkxdqj0cFQ
> > > 4. http://proceedings.mlr.press/v97/malik19a/malik19a.pdf
> > >
> > > To the best of our understanding based on these papers, the goal of the calibration approach is to clean/ remove the adversarial noises in the adversarial examples by using the knowledge extracted from clean samples. For example, people may want use auto-encoders to remove the adversarial noises; and one famous paper follows this approach is Meng&Cheng 2017 and the proposed defense in this paper has been shown not secure.  However, our goal of "buffer zones" is very different from the "calibration" approach, i.e., we want the adversarial to produce sufficient large adversarial noise to fool all our designed classifiers at the same time rather than remove the adversarial noise.
> > >
> > > Moreover, we do not see the usage of "undecided output" in these papers. It is very well appreciated if the reviewer can provide some particular references where the "undecided output" concept is used.  We also strongly believe that there may exist some papers discussing about the usage of "undecided output" because the literature of this topic is very vast.
> > >
> > >
> > > = The security of our defense is based on the diversity of classifier (using image transformations "Ax+B" and "resizing operations" as explained in Section 3 (security argument)) and majority voting trick.  This technique allows us to decide if there is any adversarial noise in the image or not. The "undecided output" raises a warning signal to the user.

---

### Decision · Program_Chairs · 2019-12-19

**Decision:**

Reject

**Comment:**

This paper formalizes the concept of buffer zones, and proposes a defense method based on a combination of deep neural networks and simple image transformations. The authors argue that the proposed method based on buffer zones is robust against state-of-the-art black box attacks methods.This paper, however, falls short of (1) unjustified claims (e.g., buffer zones are widened when the models are diverse); (2) incomplete literature survey and related work; (3) similar ideas are well-known in the literature, (4) unfair experimental evaluations and many others. Even after author response, it still does not gather support from the reviewers. Thus I recommend reject.